# Generalization Bounds via Meta-Learned Model Representations: PAC-Bayes and Sample Compression Hypernetworks

Benjamin Leblanc [1]   Mathieu Bazinet [1]   Nathaniel D'Amours [1]   Alexandre Drouin [1 2]   Pascal Germain [1]

## Abstract

Both PAC-Bayesian and Sample Compress learning frameworks have been shown instrumental for deriving tight (non-vacuous) generalization bounds for neural networks. We leverage these results in a meta-learning scheme, relying on a hypernetwork that outputs the parameters of a downstream predictor from a dataset input. The originality of our approach lies in the investigated hypernetwork architectures that encode the dataset before decoding the parameters: (1) a PAC-Bayesian encoder that expresses a posterior distribution over a latent space, (2) a Sample Compress encoder that selects a small sample of the dataset input along with a message from a discrete set, and (3) a hybrid between both approaches motivated by a new Sample Compress theorem handling continuous messages. The latter theorem exploits the pivotal information transiting at the encoder-decoder junction in order to compute generalization guarantees for each downstream predictor obtained by our meta-learning scheme.

## 1. Introduction

Machine learning is increasingly shaping our daily lives, a trend accelerated by rapid advancements in deep learning and the widespread deployment of these models across various applications (e.g., language models and AI agents). Ensuring the reliability of machine learning models is therefore more critical than ever. A fundamental aspect of reliability is understanding how well a model generalizes beyond its training data, particularly for modern neural networks, whose complexity makes it challenging to obtain strong theoretical guarantees. One way to assess generalization is through

[1]Département d'informatique et de génie logiciel, Université Laval, Québec, Canada [2]ServiceNow Research, Montréal, Canada. Correspondence to: Benjamin Leblanc <benjamin.leblanc.2@ulaval.ca>.

*Proceedings of the $42^{nd}$ International Conference on Machine Learning*, Vancouver, Canada. PMLR 267, 2025. Copyright 2025 by the author(s).

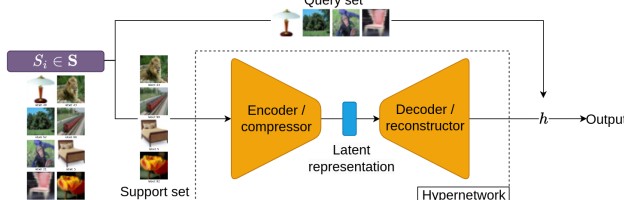

Figure 1: Overview of our meta-learning framework.

probabilistic bounds on a model's error rate. Applying these techniques to deep neural networks is challenging because classical approaches struggle to account for the true effective complexity of such models (Zhang et al., 2017), which may not be well captured by naive measures such as parameter count (Belkin et al., 2019). However, prior work suggests that if one can obtain a more compact yet expressive representation of a model's complexity, tighter generalization bounds are possible (Dziugaite et al., 2020; Wang et al., 2022; Kawaguchi et al., 2023), even when it comes to models as big as Large Langage Models (Lotfi et al., 2024a;b).

In this work, we investigate a meta-learning framework for obtaining such representations and leverage them to establish tight generalization bounds. Our approach is based on a hypernetwork architecture, trained by meta-learning, and composed of two components: an encoder that projects a set of training examples into an explicit information bottleneck (Tishby & Zaslavsky, 2015) and a decoder that generates a *downstream predictor* based on this bottleneck. We demonstrate that the complexity of this information bottleneck provides an effective measure of the downstream predictor complexity by computing generalization guarantees based on that complexity. Conceptually, our hypernetwork is akin to a learning algorithm that explicitly exposes the complexity of the models it produces. We then show how our approach can be used to obtain generalization guarantees in three theoretical frameworks: PAC-Bayesian analysis, sample compression theory, and a new hybrid approach that integrates both perspectives, each of which motivates a specific architecture for the bottleneck.

We begin by introducing the theoretical frameworks used to obtain generalization bounds (Section 2), including a new PAC-Bayes Sample Compression framework, which we

propose in Section 2.3. We then describe our meta-learning approach for training hypernetworks and how it is adapted to each theoretical framework (Section 3). Finally, we present an empirical study evaluating the quality of the obtained bounds and models on both simulated and real-world datasets (Section 4). Our results demonstrate that our approach effectively learns accurate neural network-based classifiers. We also show that the information bottleneck reliably serves as a proxy for model complexity, enabling the derivation of tight generalization guarantees.

**Related works.** The meta-learning framework was pioneered by Baxter (2000), where a learning problem encompasses multiple *tasks* under the assumption that all learning tasks are independently and identically distributed (*i.i.d.*) from a task environment distribution. Various standpoints have been considered to derive generalization guarantees in this setting, for example the VC-theory (Maurer, 2009; Maurer et al., 2016) and algorithmic stability (Maurer, 2005; Chen et al., 2020). In recent years, the PAC-Bayesian framework has been the foundation of a rich line of work: Pentina & Lampert (2014); Amit & Meir (2018); Rothfuss et al. (2021); Liu et al. (2021); Guan et al. (2022); Guan & Lu (2022); Rezazadeh (2022); Zakerinia et al. (2024). Up to our knowledge, using the sample compression framework (Littlestone & Warmuth, 1986) to derive generalization bounds for meta-learned predictors is an idea that has not been explored yet.

## 2. Learning Theory and Generalization Bounds

**The prediction problem.** A dataset $S = \{(\mathbf{x}_j, y_j)\}_{j=1}^m$ is a collection of $m$ examples, each of them being a feature-target pair $(\mathbf{x}, y) \in \mathcal{X} \times \mathcal{Y}$, and a predictor is a function $h : \mathcal{X} \to \mathcal{Y}$. We denote $\mathcal{H}$ the predictor space. Let $A$ be a learning algorithm $A : \bigcup_{k \in \mathbb{N}} (\mathcal{X} \times \mathcal{Y})^k \to \mathcal{H}$ that outputs a predictor $A(S) \in \mathcal{H}$. Given a predictor $h$ and a loss function $\ell : \mathcal{Y} \times \mathcal{Y} \to [0,1]$, the empirical loss of the predictor over a set of $m$ *i.i.d.* examples is $\widehat{\mathcal{L}}_S(h) = \frac{1}{m} \sum_{j=1}^m \ell(h(\mathbf{x}_j), y_j)$. We denote $\mathcal{D}$ the data-generating distribution over $\mathcal{X} \times \mathcal{Y}$ such that $S \sim \mathcal{D}^m$ and the generalization loss of a predictor $h$ is $\mathcal{L}_{\mathcal{D}}(h) = \mathbb{E}_{(\mathbf{x},y) \sim \mathcal{D}} [\ell(h(\mathbf{x}), y)]$.

### 2.1. PAC-Bayesian Learning Framework

The PAC-Bayes theory, initiated by McAllester (1998; 2003) and enriched by many (see Alquier (2024) for a recent survey), has become a prominent framework for obtaining non-vacuous generalization guarantees on neural network since the seminal work of Dziugaite & Roy (2017). As a defining characteristic of PAC-Bayes bounds, they rely on prior $P$ and posterior $Q$ distributions over the predictor space $\mathcal{H}$. Hence, most PAC-Bayes results are expressed as upper bounds on the $Q$-expected loss of the predictor space, thus providing guarantees on a stochastic predictor.

**Notable theoretical results.** Germain et al. (2015) expresses a general PAC-Bayesian formulation that encompasses many specific results previously stated in the literature, by resorting on a comparator function $\Delta : \mathcal{Y} \times \mathcal{Y} \to [0,1]$, used to assess the discrepancy between the expected empirical loss $\mathbb{E}_{h \sim Q} \widehat{\mathcal{L}}_S(h)$ and the expected generalization loss $\mathbb{E}_{h \sim Q} \mathcal{L}_{\mathcal{D}}(h)$. The theorem states that this discrepancy should not exceed a complexity term that notably relies on the Kullback-Leibler divergence $\mathrm{KL}(Q\|P) = \mathbb{E}_{h \sim Q} \ln \frac{Q(h)}{P(h)}$.

**Theorem 2.1** (General PAC-Bayesian theorem (Germain et al., 2015)). *For any distribution $\mathcal{D}$ on $\mathcal{X} \times \mathcal{Y}$, for any set $\mathcal{H}$ of predictors $h : \mathcal{X} \to \mathcal{Y}$, for any loss $\ell : \mathcal{Y} \times \mathcal{Y} \to [0,1]$, for any prior distribution $P$ over $\mathcal{H}$, for any $\delta \in (0,1]$ and for any convex function $\Delta : [0,1] \times [0,1] \to \mathbb{R}$, with probability at least $1 - \delta$ over the draw of $S \sim \mathcal{D}^m$, we have*

$$\forall Q \text{ over } \mathcal{H} : \Delta\Big( \mathop{\mathbb{E}}_{h \sim Q} \widehat{\mathcal{L}}_S(h), \mathop{\mathbb{E}}_{h \sim Q} \mathcal{L}_{\mathcal{D}}(h) \Big)$$
$$\leq \frac{1}{m}\Big[ \mathrm{KL}(Q\|P) + \ln\Big( \tfrac{\mathcal{I}_\Delta(m)}{\delta} \Big) \Big]$$

*with $\mathcal{I}_\Delta(m) = \mathbb{E}_{S \sim \mathcal{D}^m} \mathbb{E}_{h \sim P} e^{m\Delta(\widehat{\mathcal{L}}_S(h), \mathcal{L}_{\mathcal{D}}(h))}$.*

A common choice of comparator function is the Kullback-Leibler divergence between two Bernoulli distributions of probability of success $q$ and $p$,

$$\mathrm{kl}(q,p) = q \cdot \ln \frac{q}{p} + (1-q) \cdot \ln \frac{1-q}{1-p}, \qquad (1)$$

for which $\mathcal{I}_{\mathrm{kl}}(m) \leq 2\sqrt{m}$.

To avoid relying on a stochastic predictor, *disintegrated* PAC-Bayes theorems have been proposed to bound the loss of a single deterministic predictor, as the one of Viallard et al. (2024) reported in the appendix (Theorem E.1). They allow the study of a single hypothesis (drawn once) from the distribution $Q$.

### 2.2. Sample Compression Learning Framework

Initiated by Littlestone & Warmuth (1986) and refined by many authors over time (Attias et al., 2024; Bazinet et al., 2025; Campi & Garatti, 2023; David et al., 2016; **?**; Hanneke & Kontorovich, 2021; Hanneke et al., 2019; 2024; Laviolette et al., 2005; Marchand & Shawe-Taylor, 2002; Marchand & Sokolova, 2005; Moran & Yehudayoff, 2016; Rubinstein & Rubinstein, 2012; Shah, 2007), the sample compression theory expresses generalization bounds on predictors that rely only on a small subset of the training set, referred to as the *compression set*, valid even if the learning algorithm observes the entire training dataset. The sample compression theorems thus express the generalization ability of predictive models as an accuracy-complexity trade-off, measured respectively by the training loss and the size of the compressed representation.

**The reconstruction function.** Once a predictor $h$ is learned from a dataset $S$, *i.e.* $h = A(S)$, one can obtain

an upper bound on $\mathcal{L}_{\mathcal{D}}(h)$ thanks to the sample compression theory whenever it is possible to *reconstruct* the predictor $h$ from a compression set (that is, a subset of $S$) and an optional message (chosen from a predetermined *discrete message* set $\Omega$) with a reconstruction function $\mathcal{R} : \bigcup_{k\in\mathbb{N}}(\mathcal{X}\times\mathcal{Y})^k \times \Omega \to \mathcal{H}$. Thus, a sample compress predictor can be written

$$h = \mathcal{R}(S_{\mathbf{j}}, \omega), \qquad (2)$$

with $\mathbf{j} \subset \{i\}_{i=1}^{m}$ being the indexes of the training samples belonging to the compression set $S_{\mathbf{j}} = \{(\mathbf{x}_j, y_j)\}_{j\in\mathbf{j}}$, and $\omega \in \Omega$ being the message. In the following, we denote the set of all training indices $\mathbf{m} = \{i\}_{i=1}^{m}$, and $\mathcal{P}(\mathbf{m})$ its powerset; for compression set indices $\mathbf{j} \in \mathcal{P}(\mathbf{m})$, the complement is $\bar{\mathbf{j}} = \mathbf{m} \setminus \mathbf{j}$.

**An example: the Support Vector Machine (SVM).** Consider $A$ to be the SVM building algorithm, and $S$ a dataset with $\mathcal{Y} = \{-1, +1\}$. An easy way of reconstructing $A(S)$ is by choosing the compression set to be $S_{\mathbf{j}}$, the $m$ support vectors of $A(S)$, and having the reconstruction function to be the linear SVM algorithm $R = A$. Thus, we know that $R(S_{\mathbf{j}}) = A(S_{\mathbf{j}}) = A(S)$, without having to use any message.

**Another example: the example-aligned decision stump.** Given a dataset $S$ with $\mathcal{Y} = \{-1, +1\}$, the example-aligned decision stump (weak) learning algorithm $A$ returns a predictor $A(S) = h_{\mathbf{x}',v,k}$ such that $h_{\mathbf{x}',v,k}(\mathbf{x}) = 1$ if $v(x_k - x'_k) \leq 0$ or $-1$ otherwise, for some datapoint $\mathbf{x}' = (x'_1, \ldots, x'_d) \in S$, direction $v \in \{-1, +1\}$ and index $k \in \{1, \ldots, d\}$. We can fully reconstruct the stump with $R(S_{\mathbf{j}}, v) = h_{\mathbf{x}',v,k}$, where $S_{\mathbf{j}} = \{(\mathbf{x}', y')\}$ is the compression set and $\omega = (v, k)$ is the message.

**Notable theoretical results.** Theorem 2.2 below, due to Laviolette et al. (2005), improves the bound developed for their Set Covering Machine (SCM) algorithm (Marchand & Shawe-Taylor, 2001; Marchand & Shawe-Taylor, 2002; Marchand & Sokolova, 2005). It is premised on two data-independent distributions: $P_{\mathcal{P}(\mathbf{m})}$ on the compression set indices $\mathcal{P}(\mathbf{m})$, and $P_\Omega$ on a discrete set of messages $\Omega$. Noteworthy, the bound is valid solely for binary-valued losses, as it considers each "successful" and "unsuccessful" prediction to be the result of a Bernoulli distribution.

**Theorem 2.2** (Sample compression - binary loss (Laviolette et al., 2005)). *For any distribution $\mathcal{D}$ over $\mathcal{X}\times\mathcal{Y}$, for any set $J \subseteq \mathcal{P}(\mathbf{m})$, for any distribution $P_J$ over $J$, for any $P_\Omega$ over $\Omega$, for any reconstruction function $\mathcal{R}$, for any binary loss $\ell : \mathcal{Y}\times\mathcal{Y} \to \{0,1\}$ and for any $\delta \in (0,1]$, with probability at least $1-\delta$ over the draw of $S \sim \mathcal{D}^m$, we have*

$$\forall \mathbf{j} \in J, \omega \in \Omega : \mathcal{L}_{\mathcal{D}}\big(\mathcal{R}(S_{\mathbf{j}}, \omega)\big) \leq$$
$$\underset{r\in[0,1]}{\mathrm{argsup}} \left\{ \sum_{k=1}^{K} \binom{|\bar{\mathbf{j}}|}{k} r^k (1-r)^{|\bar{\mathbf{j}}|-k} \geq P_J(\mathbf{j}) \, P_\Omega(\omega) \, \delta \right\},$$

*with* $K = |\bar{\mathbf{j}}| \widehat{\mathcal{L}}_{S_{\bar{\mathbf{j}}}}\big(\mathcal{R}(S_{\mathbf{j}}, \omega)\big)$.

Theorem 2.2 is limited in its scope, for many tasks involve non-binary losses (e.g. regression tasks, cost-sensitive classification). The following recent result, due to Bazinet et al. (2025), permits real-valued losses $\ell : \mathcal{Y}\times\mathcal{Y} \to \mathbb{R}$. Given a *comparator function* $\Delta$ analogous to the one classically used in PAC-Bayesian theorems, Theorem 2.3 bounds the discrepancy between the empirical loss of the reconstructed hypothesis $\mathcal{R}(S_{\mathbf{j}}, \omega)$ on the complement set $S_{\bar{\mathbf{j}}}$ and its generalization loss on the data distribution $\mathcal{D}$.

**Theorem 2.3** (Sample compression - real-valued losses (Bazinet et al., 2025)). *For any distribution $\mathcal{D}$ over $\mathcal{X}\times\mathcal{Y}$, for any set $J \subseteq \mathcal{P}(\mathbf{m})$, for any distribution $P_J$ over $J$, for any distribution $P_\Omega$ over $\Omega$, for any reconstruction function $\mathcal{R}$, for any loss $\ell : \mathcal{Y}\times\mathcal{Y} \to \mathbb{R}$, for any function $\Delta : \mathbb{R}\times\mathbb{R} \to \mathbb{R}$ and for any $\delta \in (0,1]$, with probability at least $1-\delta$ over the draw of $S \sim \mathcal{D}^m$, we have*

$$\forall \mathbf{j} \in J, \omega \in \Omega : \Delta\Big(\widehat{\mathcal{L}}_{S_{\bar{\mathbf{j}}}}(\mathcal{R}(S_{\mathbf{j}}, \omega)), \mathcal{L}_{\mathcal{D}}(\mathcal{R}(S_{\mathbf{j}}, \omega))\Big)$$
$$\leq \frac{1}{m - |\mathbf{j}|} \left[ \ln\left( \frac{\mathcal{J}_\Delta(\mathbf{j}, \omega)}{P_J(\mathbf{j}) \cdot P_\Omega(\omega) \cdot \delta} \right) \right],$$

*with*

$$\mathcal{J}_\Delta(\mathbf{j}, \omega) = \underset{T_{\mathbf{j}}\sim\mathcal{D}^{|\mathbf{j}|}}{\mathbb{E}} \underset{T_{\bar{\mathbf{j}}}\sim\mathcal{D}^{m-|\bar{\mathbf{j}}|}}{\mathbb{E}} e^{|\bar{\mathbf{j}}|\Delta\Big(\widehat{\mathcal{L}}_{T_{\bar{\mathbf{j}}}}(\mathcal{R}(T_{\mathbf{j}}, \omega)), \mathcal{L}_{\mathcal{D}}(\mathcal{R}(T_{\mathbf{j}}, \omega))\Big)}.$$

In order to compute a numerical bound on the generalization loss $\mathcal{L}_{\mathcal{D}}(\mathcal{R}(S_{\mathbf{j}}, \omega))$, one must commit to a choice of $\Delta$. See Appendix A for corollaries involving specific choices of comparator function. In particular, the choice of comparator function $\mathrm{kl}(q, p)$ of Equation (1) leads to $\mathcal{J}_{\mathrm{kl}}(\mathbf{j}, \omega) \leq 2\sqrt{m - |\mathbf{j}|}$ (see Corollary A.2).

### 2.3. A New PAC-Bayes Sample Compression Framework

Our first contribution lies in the extension of Theorem 2.3 to *real-valued messages*. This is achieved by using a strategy from the PAC-Bayesian theory: we consider a data-independent prior distribution over the messages $\Omega$, denoted $P_\Omega$, and a data-dependent posterior distribution over the messages, denoted $Q_\Omega$. We then obtain a bound for the expected loss over $Q_\Omega$. Note that this new result shares similarities with the existing PAC-Bayes sample compression theory (Germain et al., 2011; 2015; Laviolette & Marchand, 2005; ?), which gives PAC-Bayesian bounds for an expectation of data-dependent predictors given distributions on both the compression set and the messages. Our result differs by restricting the expectation solely according to the message.

**Theorem 2.4** (PAC-Bayes Sample compression - real-valued losses with continuous messages). *For any distribution $\mathcal{D}$ over $\mathcal{X}\times\mathcal{Y}$, for any set $J \subseteq \mathcal{P}(\mathbf{m})$, for any distribution $P_J$ over $J$, for any prior distribution $P_\Omega$ over $\Omega$, for any reconstruction function $\mathcal{R}$, for any loss $\ell : \mathcal{Y}\times\mathcal{Y} \to [0,1]$, for any convex function $\Delta : [0,1]\times[0,1] \to \mathbb{R}$ and for any*

$\delta \in (0, 1]$, *with probability at least* $1 - \delta$ *over the draw of* $S \sim \mathcal{D}^m$, *we have*

$$\forall \mathbf{j} \in J, Q_\Omega \, over \, \Omega :$$

$$\Delta \left( \mathop{\mathbb{E}}_{\omega \sim Q_\Omega} \widehat{\mathcal{L}}_{S_{\bar{\mathbf{j}}}} \left( \mathcal{R}(S_{\mathbf{j}}, \omega) \right), \mathop{\mathbb{E}}_{\omega \sim Q_\Omega} \mathcal{L}_\mathcal{D} \left( \mathcal{R}(S_{\mathbf{j}}, \omega) \right) \right)$$

$$\leq \frac{1}{m - \max_{\mathbf{j} \in J} |\mathbf{j}|} \left[ \mathrm{KL}(Q_\Omega || P_\Omega) + \ln \left( \frac{\mathcal{A}_\Delta(m)}{P_J(\mathbf{j})\delta} \right) \right]$$

*with*

$$\mathcal{A}_\Delta(m) = \mathop{\mathbb{E}}_{(\mathbf{j}, \omega) \sim P} \mathop{\mathbb{E}}_{T \sim \mathcal{D}^m} e^{|\bar{\mathbf{j}}| \Delta \left( \widehat{\mathcal{L}}_{T_{\bar{\mathbf{j}}}}(\mathcal{R}(T_{\mathbf{j}}, \omega)), \mathcal{L}_\mathcal{D}(\mathcal{R}(T_{\mathbf{j}}, \omega)) \right)}.$$

See Appendix D for the complete proof of Theorem 2.4 and specific choices of $\Delta$. In particular, $\mathcal{A}_{\mathrm{kl}}(m) \leq \mathbb{E}_{\mathbf{j} \sim P_J} 2\sqrt{m - |\mathbf{j}|} \leq 2\sqrt{m - \min_{\mathbf{j} \in J} |\mathbf{j}|}$ (Cor. D.2). Note that, in the setting where the compression set size is fixed (*i.e.*, $\forall \mathbf{j} \in J$, $|\mathbf{j}| = c$), Theorem 2.3 is a particular case of Theorem 2.4 where the message space $\Omega$ is discrete and the posterior $Q_\Omega$ is a Dirac over a single $\omega^\star \in \Omega$ (*i.e.*, $Q_\Omega(\omega^\star) = 1$).

Leveraging the PAC-Bayes *disintegrated* theorem of Viallard et al. (2024), we also obtain a variant of Theorem 2.4 valid for a single deterministic predictor $\mathcal{R}(S_{\mathbf{j}^\star}, \omega^\star)$ (with $\omega^\star$ drawn once according to $Q_\Omega$), instead of the expected loss according to $Q_\Omega$. This new result requires the use of a singular compression function $\zeta(S, P_\Omega) = (S_{\mathbf{j}^\star}, Q_\Omega^S)$, which takes as input a dataset and a prior distribution over the messages, and outputs a compression set and a posterior distribution.

**Theorem 2.5** (Disintegrated PAC-Bayes Sample compression bound). *For any distribution $\mathcal{D}$ over $\mathcal{X} \times \mathcal{Y}$, for any set $J \subseteq \mathcal{P}(\mathbf{m})$, for any prior distribution $P_J$ over $J$, for any prior distribution $P_\Omega$ over $\Omega$, for any reconstruction function $\mathcal{R}$, for any compression function $\zeta$, for any loss $\ell : \mathcal{Y} \times \mathcal{Y} \to [0, 1]$, for any $\alpha > 1$, for any convex function $\Delta : [0, 1] \times [0, 1] \to \mathbb{R}$, for any $\delta \in (0, 1]$, with probability at least $1 - \delta$ over the draw of $S \sim \mathcal{D}^m$ (which leads to $(S_{\mathbf{j}^\star}, Q_\Omega^S) = \zeta(S, P_\Omega)$ ), and $\omega^\star \sim Q_\Omega^S$, we have*

$$(m - \max_{\mathbf{j} \in J} |\mathbf{j}|) \Delta \left( \widehat{\mathcal{L}}_{S_{\mathbf{j}^\star}} \left( \mathcal{R}(S_{\mathbf{j}^\star}, \omega^\star) \right), \mathcal{L}_\mathcal{D} \left( \mathcal{R}(S_{\mathbf{j}^\star}, \omega^\star) \right) \right)$$

$$\leq \frac{2\alpha - 1}{\alpha - 1} \ln \frac{2}{\delta} + \ln \frac{1}{P_J(\mathbf{j}^\star)} + D_\alpha(Q_\Omega^S || P_\Omega) + \ln \mathcal{A}_\Delta(m).$$

The proof of Theorem 2.5 is given in Appendix E.

# 3. Meta-Learning with Hypernetworks

**The meta-prediction problem.** We now introduce the meta-learning setting that we leverage in order to benefit from the generalization guarantees presented in the previous section.

Let a *task* $\mathcal{D}_i$ be a realization of a meta distribution $\mathbf{D}$, and $S_i \sim \mathcal{D}_i^{m_i}$ be a dataset of $m_i$ *i.i.d.* samples from a given task.

A meta-learning algorithm receives as input a meta-dataset $\mathbf{S} = \{S_i\}_{i=1}^n$, that is a collection of $n$ datasets obtained from distributions $\{\mathcal{D}_i\}_{i=1}^n$. The aim is to learn a meta-predictor $\mathcal{H}$ that, given only a few sample $S' \sim (\mathcal{D}')^{m'}$ for a new task $\mathcal{D}' \sim \mathbf{D}$, produces a predictor $h' = \mathcal{H}(S')$ that generalizes well, i.e., with low generalization loss $\mathcal{L}_{\mathcal{D}'}(h')$.

In conformity with classical meta-learning literature (Setlur et al., 2020; Vinyals et al., 2016), the following considers that each task dataset $S_i \in \mathbf{S}$ is split into a *support set* $\hat{S}_i \subset S_i$ and a *query set* $\hat{T}_i = S_i \backslash \hat{S}_i$; the former is used to learn the predictor $h_i = \mathcal{H}(\hat{S}_i)$ and the latter to estimate $h_i$'s loss:

$$\widehat{\mathcal{L}}_{\hat{T}_i}(h_i) = \frac{1}{|\hat{T}_i|} \sum_{(\mathbf{x}, y) \in \hat{T}_i} \ell \left( h_i(\mathbf{x}), y \right).$$

**Meta-learning hypernetworks.** We propose to use a hypernetwork as meta-predictor, that is a neural network $\mathcal{H}_\theta$ with parameters $\theta$, whose output is an array $\gamma \in \mathbb{R}^{|\gamma|}$ that is in turn the parameters of a *downstream network* $h_\gamma : \mathbb{R}^d \to \mathcal{Y}$. The particularity of our approaches is the use of an explicit bottleneck in the hypernetwork architecture. An overview is given in Figure 1. Hence, given a training set $S$, we train the hypernetwork by optimizing its parameters $\theta$ in order to minimize the empirical loss of the downstream predictor $h_\gamma$ on the query set. That is, given a training meta-dataset $\mathbf{S} = \{S_i\}_{i=1}^n$, we propose to optimize the following objective.

$$\min_\theta \left\{ \frac{1}{n} \sum_{i=1}^n \widehat{\mathcal{L}}_{\hat{T}_i}(h_{\gamma_i}) \, \middle| \, \gamma_i = \mathcal{H}_\theta(\hat{S}_i) \right\}. \qquad (3)$$

**Toward generalization bounds.** From now on, our aim is to study variants of hypernetwork architectures for the meta-predictor $\mathcal{H}_\theta$, each variant inspired by the learning frameworks of Section 2. By doing so, once $\mathcal{H}_\theta$ is learned, every downstream network $h_{\gamma'} = \mathcal{H}_\theta(S')$ comes with its own risk certificate, *i.e.*, an upper bound on its generalization loss $\mathcal{L}_{\mathcal{D}'}(h_{\gamma'})$ statistically valid with high probability.

We stress that, while the usual meta-learning bounds are computed after the meta-learning training phase in order to guarantee the expected performance of future downstream predictors learned on yet unseen tasks, the bounds we provide concern the generalization of downstream predictors once they are outputted by the meta-predictor.

## 3.1. PAC-Bayes Encoder-Decoder Hypernetworks

We depart from previous works on PAC-Bayesian meta-learning and from classical PAC-Bayes, which consider a hierarchy of (meta-)prior and (meta-)posterior distributions (*e.g.*, Pentina & Lampert, 2014). Instead, we consider distributions over a latent representation learned by the hypernetwork $\mathcal{H}_\theta$. To encourage this representation to focus on relevant information, we adopt an encoder-decoder architecture $\mathcal{H}_\theta(\cdot) = \mathcal{D}_\psi(\mathcal{E}_\phi(\cdot))$, with $\theta = \{\phi, \psi\}$. The encoder $\mathcal{E}_\phi$

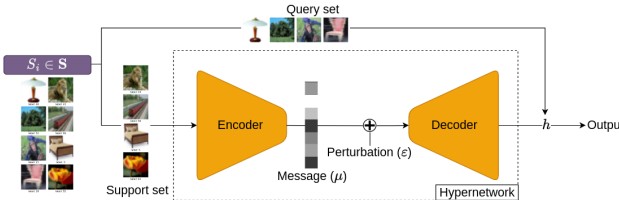

Figure 2: The PAC-Bayes hypernetwork.

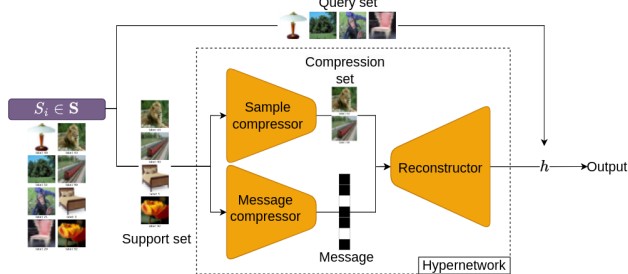

Figure 3: The Sample Compression hypernetwork.

*compresses* the relevant dataset information into a vector $\boldsymbol{\mu} \in \mathbb{R}^{|\boldsymbol{\mu}|}$ (typically, $|\boldsymbol{\mu}| \ll |\gamma|$). The vector $\boldsymbol{\mu}$ is then treated as the mean of the Gaussian posterior distribution $Q_{\boldsymbol{\mu}} = \mathcal{N}(\boldsymbol{\mu}, \mathbf{I})$ over the latent representation space, such that the decoder $\mathcal{D}_\psi$ maps the realizations drawn from $Q_{\boldsymbol{\mu}}$ to the downstream parameters $\gamma$. Figure 2 illustrates this architecture.

**Training objective.** Based on the above, learning the proposed encoder-decoder hypernetwork amount to solve

$$\min_{\psi,\phi} \left\{ \frac{1}{n} \sum_{i=1}^n \mathbb{E}\, \widehat{\mathcal{L}}_{\hat{T}_i}(h_{\gamma_i}) \,\middle|\, \gamma_i = \mathcal{D}_\psi(\boldsymbol{\mu}_i + \boldsymbol{\epsilon}); \, \boldsymbol{\mu}_i = \mathcal{E}_\phi(\hat{S}_i) \right\},$$

with $\boldsymbol{\epsilon} \sim \mathcal{N}(\mathbf{0}, \mathbf{I})$.

**Bound computation.** Given a new task sample $S' \sim (\mathcal{D}')^{m'}$, we obtain from the PAC-Bayesian Theorem 2.1 (using a prior $P_0 = \mathcal{N}(\mathbf{0}, \mathbf{I})$ over the latent representation space and the comparator function of Equation (1)), the following upper bound on the expected loss according to $\mathcal{D}'$:

$$\tau^* = \operatorname*{argsup}_{\tau \in [0,1]} \left\{ \mathrm{kl}\Big(\mathbb{E}\, \widehat{\mathcal{L}}_{S'}(h_{\gamma'}), \tau\Big) \leq \frac{\frac{1}{2}\|\boldsymbol{\mu}\|^2 + \ln\frac{2\sqrt{m'}}{\delta}}{m'} \right\},$$
(4)

where $\boldsymbol{\mu}$ is the mean of the Gaussian posterior distribution. That is, with probability at least $1 - \delta$, we have $\mathbb{E}\, \mathcal{L}_{\mathcal{D}'}(h_{\gamma'}) \leq \tau^*$, where the expectation comes from the stochastic latent space.

### 3.2. Sample Compression Hypernetworks

Let us now design a hypernetwork architecture derived from the sample compression theory presented in Section 2.2. Similar to the previously presented PAC-Bayes encoder-decoder, the architecture detailed below acts as an *information bottleneck*. However, instead of a PAC-Bayesian encoder $\mathcal{E}_\phi$ mapping the dataset to a latent representation, we consider a *sample compressor* $\mathcal{C}_{\phi_1}$ that selects a few samples from the training set. These become the input of a *reconstructor* $\mathcal{R}_\psi$ that produces the parameters of a downstream predictor, akin to the decoder in our PAC-Bayesian approach. In line with the sample compress framework, our *reconstructor* $\mathcal{R}_\psi$ optionally takes additional message input, given by a *message compressor* $\mathcal{M}_{\phi_2}$. This amounts to *learn the reconstruction function*; an idea that has not been envisaged

before in the sample compress literature (to our knowledge). The overall resulting architecture is illustrated by Figure 3.

**Reconstructor hypernetwork.** In line with the sample compression framework of Section 2.2, our *reconstructor* $\mathcal{R}_\psi$ takes two complementary inputs:

1. A compression set $S_{\mathbf{j}}$ containing a fixed number of $c$ examples;

2. (optionally) A message $\boldsymbol{\omega} \in \{-1, 1\}^b = \Omega$, that is a binary-valued vector of size $b$.

The output of the reconstructor hypernetwork is an array $\gamma \in \mathbb{R}^{|\gamma|}$ that is in turn the parameters of a *downstream network* $h_\gamma : \mathbb{R}^d \to \mathcal{Y}$. Hence, given a (single task) training set $S$, a compression set $S_{\mathbf{j}} \subset S$ and a message $\boldsymbol{\omega} \in \Omega$, a reconstructor is trained by optimizing its parameters $\psi$ in order to minimize the empirical loss of the downstream predictor $h_\gamma$ on the complement set $S_{\bar{\mathbf{j}}} = S \setminus S_{\mathbf{j}}$:

$$\min_\psi \left\{ \frac{1}{m - |\mathbf{j}|} \sum_{(\mathbf{x},y) \in S_{\bar{\mathbf{j}}}} \ell\big(h_\gamma(\mathbf{x}), y\big) \,\middle|\, \gamma = \mathcal{R}_\psi(S_{\mathbf{j}}, \boldsymbol{\omega}) \right\}. \quad (5)$$

Note that the above corresponds to the minimization of the empirical loss term $\widehat{\mathcal{L}}_{S_{\bar{\mathbf{j}}}}(\cdot)$ of the sample compression bounds (*e.g.*, Theorem 2.3). However, to be statistically valid, these bounds must not be computed on the same data used to learn the reconstructor. Fortunately, our meta-learning framework satisfies this requirement since the reconstructor is learned on the meta-training set, rather than the task of interest. Note that the compression set and the message are not *given*, but outputted by $\mathcal{C}_{\phi_1}$ and $\mathcal{M}_{\phi_2}$.

**Training objective.** Our goal is to learn parameters $\psi$, $\phi_1$ and $\phi_2$ such that, for any task $\mathcal{D}' \sim \mathbf{D}$ producing $S' \sim (\mathcal{D}')^{m'}$, the resulting output gives rise to a downstream predictor $h_{\gamma'}$ of low generalization loss $\mathcal{L}_{\mathcal{D}'}(h_{\gamma'})$, with

$$\gamma' = \mathcal{R}_\psi\big(\mathcal{C}_{\phi_1}(S'), \mathcal{M}_{\phi_2}(S')\big). \quad (6)$$

Given a training meta-dataset $\mathbf{S} = \{S_i\}_{i=1}^n$, we propose to

optimize the following objective:

$$\min_{\psi,\phi_1,\phi_2}\left\{\frac{1}{n}\sum_{i=1}^{n}\widehat{\mathcal{L}}_{\hat{T}_i}(h_{\gamma_i})\,\middle|\,\gamma_i=\mathcal{R}_\psi\Big(\mathcal{C}_{\phi_1}(\hat{S}_i),\mathcal{M}_{\phi_2}(\hat{S}_i)\Big)\right\}.\ (7)$$

Note that the learning objective is a surrogate for Equation (5), as the complement of the compression set $S_{\bar{\mathbf{j}}}$ is replaced by the query set $\hat{T}_i$ in Equation (7). The pseudocode of the proposed approach can be found in Appendix G.

**Bound computation.** When the 0-1 loss is used, the generalization bound from Theorem 2.2 is computed, using a fixed size $c$ for the compression sets; given a dataset size $m'$, we use $J=\{\mathbf{j}\in\mathcal{P}(\mathbf{m'})\ :\ |\mathbf{j}|=c\}$ and a uniform probability distribution over all distinct compression sets (sets that are not permutations of one another): $p=P_J(\mathbf{j})=\binom{m'}{c}^{-1}\ \forall\mathbf{j}\in J$. When using the optional message compressor module, we set a message size $b$ and a uniform distribution over all messages of size $b$: $P_\Omega(\boldsymbol{\omega})=2^{-b}\ \forall\boldsymbol{\omega}\in\{-1,1\}^b$, leading to the following upper bound on the loss:

$$\tau^*=\operatorname*{argsup}_{\tau\in[0,1]}\left\{\sum_{k=1}^{K}\binom{m'-c}{k}\tau^k(1-\tau)^{m'-c-k}\ge p\,2^{-b}\,\delta\right\},$$

with $K=|\bar{\mathbf{j}}|\widehat{\mathcal{L}}_{S'_{\bar{\mathbf{j}}}}(h_{\gamma'})$.

When a real-valued loss is used, applying Theorem 2.3 with the comparator function of Equation (1), we obtain the following upper bound on the loss:

$$\tau^*=\operatorname*{argsup}_{\tau\in[0,1]}\left\{\mathrm{kl}\Big(\widehat{\mathcal{L}}_{S'_{\bar{\mathbf{j}}}}(h_{\gamma'}),\tau\Big)\le\frac{1}{m'-c}\ln\frac{2^{b+1}\sqrt{m'-c}}{p\,\delta}\right\}.$$

That is, $\mathcal{L}_{\mathcal{D}'}(h_{\gamma'})\le\tau^*$ with probability at least $1-\delta$.

### 3.3. PAC-Bayes Sample Compression Hypernetworks

As a third hypernetwork architecture, the new theoretical perspective presented in Section 2.3 led to a hybrid between previous PAC-Bayesian and sample compression approaches.

Recall that Theorem 2.4 is obtained by handling the message of the sample compress framework in a PAC-Bayesian fashion, enabling the use of a posterior distribution over continous messages. Hence, this motivates revisiting the sample compress hypernetwork of Section 3.2 by replacing the message compressor $\mathcal{M}_{\phi_2}$ (outputting a binary vector) by the PAC-Bayes encoder of Section 3.1. We denote the latter $\mathcal{E}_{\phi_2}$, whose task is to output the mean $\boldsymbol{\mu}\in\mathbb{R}^b$ of a posterior distribution $Q_{\Omega,\boldsymbol{\mu}}=\mathcal{N}(\boldsymbol{\mu},\mathbf{I})$ over a real-valued message space $\Omega=\mathbb{R}^b$. The sample compressor remaining unchanged from Section 3.2, the *PAC-Bayes Sample Compress architecture* is expressed by $\mathcal{H}_\theta(\cdot)=\mathcal{R}_\psi(\mathcal{C}_{\phi_1}(\cdot),\mathcal{E}_{\phi_2}(\cdot))$. Figure 4 illustrates the resulting architecture.

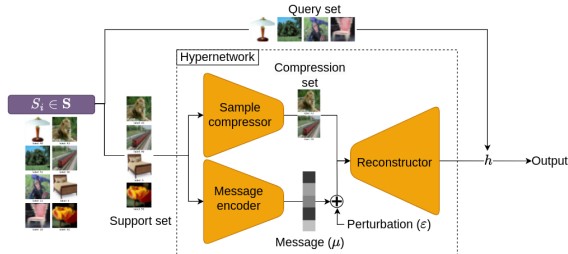

Figure 4: The PAC-Bayes Sample Compression hypernetwork.

**Training objective.** Based on the above formulation, we obtain the following training objective:

$$\min_{\psi,\phi_1,\phi_2}\left\{\frac{1}{n}\sum_{i=1}^{n}\mathbb{E}\,\widehat{\mathcal{L}}_{\hat{T}_i}(h_{\gamma_i})\,\middle|\,\gamma_i=\mathcal{R}_\psi\big(\mathcal{C}_{\phi_1}(\hat{S}_i),\mathcal{E}_{\phi_2}(\hat{S}_i)+\boldsymbol{\epsilon}\big)\right\},$$

with $\boldsymbol{\epsilon}\sim\mathcal{N}(\mathbf{0},\mathbf{I})$.

**Bound computation.** From Theorem 2.4, using a fixed compression set size $c$, a prior $P_{\Omega,\mathbf{0}}=\mathcal{N}(\mathbf{0},\mathbf{I})$ over the real-valued message space of size $b$, a uniform probability over the compression set choice $p=P_J(\mathbf{j})=\binom{m'}{c}^{-1}\ \forall\mathbf{j}\in J$, and the comparator function of Equation (1), we obtain the following upper bound on the expected loss:

$$\tau^*=\operatorname*{argsup}_{\tau\in[0,1]}\left\{\mathrm{kl}\Big(\mathbb{E}\,\widehat{\mathcal{L}}_{S'_{\bar{\mathbf{j}}}}(h_{\gamma'}),\tau\Big)\le\frac{\frac{1}{2}\|\boldsymbol{\mu}'\|^2+\ln\frac{2\sqrt{m'-c}}{p\cdot\delta}}{m'-c}\right\}.$$

That is, with probability at least $1-\delta$, we have $\mathbb{E}\,\mathcal{L}_{\mathcal{D}'}(h_{\gamma'})\le\tau^*$, where the expectation comes from the stochastic message treatment.

To compute the disintegrated bound of Theorem 2.5, we choose $\zeta$ to output the compression set $S'_{\mathbf{j}}=\mathcal{C}_{\phi_1}(S')$ and a normal distribution centered on the message $\mathcal{E}_{\phi_2}(S')$, denoted $Q^S_\Omega=\mathcal{N}(\mathcal{E}_{\phi_2}(S'),\mathbf{I})$. Thus, we have $\zeta(S',P_\Omega)=(S'_{\mathbf{j}},Q^S_\Omega)$. After sampling a message $\omega\sim Q^S_\Omega$, we define $\gamma'=\mathcal{R}_\psi(S'_{\mathbf{j}},\omega)$. Applying Theorem 2.5 with $\alpha=2$, we have the following upper bound on the loss :

$$\tau^*=\operatorname*{argsup}_{\tau\in[0,1]}\left\{\mathrm{kl}\Big(\widehat{\mathcal{L}}_{S'_{\mathbf{j}}}(h_{\gamma'}),\tau\Big)\le\frac{\|\boldsymbol{\mu}'\|^2+\ln\frac{16\sqrt{m'-c}}{p\cdot\delta^3}}{m'-c}\right\}.$$

That is, $\mathcal{L}_{\mathcal{D}'}(h_{\gamma'})\le\tau^*$ with probability at least $1-\delta$.

## 4. Experiments

We now study the performance of models learned using our meta-learning framework as well as the quality of the obtained bounds. Then, we report results on a synthetic meta-learning task (Section 4.2) and two real-world meta-learning tasks (Sections 4.3 and 4.4).

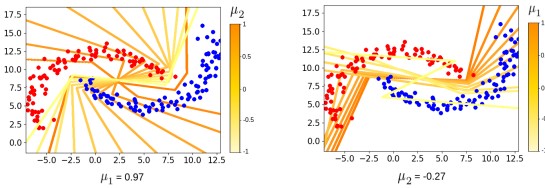

Figure 5: Illustration of decision boundaries generated by downstream predictors $h_\gamma$, when parameters $\gamma$ are given by a PAC-Bayes decoder $\mathcal{D}_\psi(\boldsymbol{\mu})$ with input $\boldsymbol{\mu} = (\mu_1, \mu_2) \in \mathbb{R}^2$. The encoder generated the message $(0.97, -0.27)$ on shown training datapoints. **Left:** The first latent dimension is fixed ($\mu_1 = 0.97$), while the second varies ($\mu_2 \in [-1, 1]$). **Right:** The first dimension varies ($\mu_1 \in [-1, 1]$), while the second is fixed $\mu_2 = -0.27$.

### 4.1. Implementation Details

In each task, we split our training datasets into train and validation datasets; for each meta-learning hypernetwork, the hyperparameters are selected according to the error made on the validation datasets. Detailed hyperparameters used for each experiment are given in Appendix H.[1]

**DeepSet dataset encoding.** The hypernetwork $\mathcal{H}_\theta(S)$ must be invariant to the permutation of its input $S$: the order of the examples in the input dataset should not affect the resulting encoding. Modules such as FSPool (Zhang et al., 2020) or a transformer (Vaswani et al., 2017) ensure such property. Our experiments use a simpler mechanism that is inspired by the DeepSet module (Zaheer et al., 2017).

**Definition 4.1** (DeepSet Module)**.** For binary tasks: given a data-matrix $\mathbf{X} \in \mathbb{R}^{m \times d}$ and a binary label vector $\mathbf{y} \in \{-1, 1\}^m$, the output of a *DeepSet module* is the embedding $\mathbf{z} \in \mathbb{R}^{d'}$, obtained by first applying a fully-connected neural network $g_\omega : \mathbb{R}^d \to \mathbb{R}^{d'}$ to each row of $\mathbf{X}$, sharing the weights across rows, to obtain a matrix $\mathbf{M} \in \mathbb{R}^{m \times d'}$ and then aggregating the result column-wise: $\mathbf{z} = \frac{1}{m}\mathbf{M}^T\mathbf{y}$.

For $\kappa$ class tasks, where $\kappa > 2$: given a data-matrix $\mathbf{X} \in \mathbb{R}^{m \times d}$ and a one-hot encoding of the label $\mathbf{Y}$, the output of a *DeepSet module* is the embedding $\mathbf{z} \in \mathbb{R}^{d'}$, obtained by first applying a fully-connected neural network $g_\omega : \mathbb{R}^d \to \mathbb{R}^{d'+\kappa}$ to each row of $\mathbf{X}$, where the label representation have been appended, to obtain a matrix $\mathbf{M} \in \mathbb{R}^{m \times (d'+\kappa)}$ and then aggregating the result column-wise: $\mathbf{z} = \frac{1}{m}\mathbf{M}^T\mathbf{1}$.

**PAC-Bayes encoder / message compressor.** The PAC-Bayes encoder takes as input a dataset and outputs a continuous representation.[2] It is composed of a DeepSet module, followed by a feedforward network. Its last activation function is a Tanh function, so that the message is close to $\mathbf{0}$, leading to a bound that is not penalized much from

---

[1]Our code is available at https://github.com/GRAAL-Research/DeepRM.

[2]In a slight language abuse, from now on, this continuous representation will be referred to as *message*, just like the output of the message compressor.

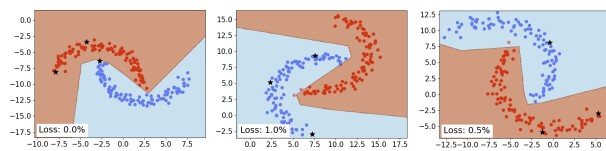

Figure 6: Examples of decision boundaries given by the downstream predictors, with a compression set of size 3 and without message, on test datasets. The stars show the retained points from the sample compressor $\mathcal{C}_{\phi_1}$. As shown by the axes, each plot is centered and scaled on the moons datapoints.

having an important representation size (see Equation (4)).

As for the message compressor, it has the same architecture as the PAC-Bayes encoder, but its final activation is the *sign* function coupled with the straight-through estimator (Hinton, 2012) in order to generate binary values.

**Sample compressor.** Given a fixed compression set size $c$, the sample compressor $\mathcal{C}_{\phi_1}$ is composed of $c$ independent attention mechanisms (Bahdanau et al., 2015). The queries are the result of a DeepSet module (see Definition 4.1), the keys are the result of a fully-connected neural network and the values are the features themselves. Each attention mechanism outputs a probability distribution over the examples indices from the support set, and the example having the highest probability is added to the compression set.

**Decoder / reconstructor.** The input of the decoder / reconstructor is passed through a DeepSet. Then, both the obtained compression set embedding and the message (if there is one) are fed to a feedforward neural network, whose output constitutes the parameters of the downstream network.

**Nomenclature.** In the following, we refer to our different meta-predictors as such: PAC-Bayes Hypernetwork (**PBH**); Sample Compression Hypernetwork, without messages (**SCH$_-$**) and with messages (**SCH$_+$**); and PAC-Bayes Sample Compression Hypernetwork (**PB SCH**).

### 4.2. Numerical Results on a Synthetic Problem

We first conduct an experiment on the *moons* 2-D synthetic dataset from Scikit-learn (Pedregosa et al., 2011), which consists of two interleaving half circles with small Gaussian noise, the goal being to better understand the inner workings of the proposed approach. We generate tasks by rotating (random degree in $[0, 360]$), translating (random moon center in $[-10, 10]^2$), and re-scaling the moons (random scaling factor in $[0.2, 5]$). The *moons* meta-train set consists of 300 tasks of 200 examples, while the meta-test set consists of 100 tasks of 200 examples. We randomly split each dataset into support and query of equal size. See Appendix H for implementation details.

Figure 5 displays the decision boundaries of a predictor trained with the PBH model, given a message of size $|\boldsymbol{\mu}| = 2$,

on a random test dataset. We plot the result for many values of the message, displaying its effect on the decision boundary of the resulting downstream predictor; we observe that each dimension of the message encapsulates a unique piece of information about the task at hand.

Figure 6 displays the decision boundaries of a predictor (trained with the SCH_ meta-predictor, with a compression set of size $c = 3$) for three different test tasks. We see that the sample compressor selects three examples far from each other, efficiently *compressing* the task, while the reconstructor correctly leverages the information contained in these examples to correctly parameterize the downstream predictor, leading to an almost perfect classification. Recall that SCH_ does not incorporate any message.

### 4.3. Test Case: Noisy MNIST

Following Amit & Meir (2018), we experiment with three different yet related task environments, based on augmentations of the MNIST dataset (LeCun et al., 1998). In each environment, each classification task is created by the random permutation of a given number (100, 200, and 300) of pixels. The pixel permutations are created by a limited number of location swaps to ensure that the tasks stay reasonably related in each environment. In each of the three experiments, the meta-training set consists of 10 tasks of 60'000 training examples, while the meta-test set consists of 20 tasks of 2000 examples.

We compare our approaches to algorithms yielding PAC-Bayesian bounds as benchmarks. The reported bounds concern the generalization property of the trained model on a given test task. We chose a fully connected network with no more than 3 hidden layers and a linear output layer as a backbone, as per the selected benchmarks. We also report the performances of a strawman: an *opaque encoder* outputting nothing, followed by a decoder which input is a predefined sole constant scalar; though the reconstructor can be trained, the hypernetwork always generates the same predictor, no matter the input. We report the test bounds and test errors on the novel tasks of various methods in Table 1. More details about the experiment setup can be found in Appendix H.

When it comes to generalization bounds, our approaches outperform the benchmarks. However, our approaches cannot learn from the task environment, their performances being similar to those of the strawman. This is because the difference between the various tasks is very subtle; such a setting is not well-suited for our approaches, which rely on the encapsulation of the differences between tasks in a message or a compression set. The benchmark methods perform well because the posterior (over all of the downstream predictor), for each test task, is *similar* to the prior. Indeed, the various tasks do not vary much; this is reflected by a single predictor (the strawman, an opaque hypernetwork) yielding competitive performances across all of the test tasks.

This matter is empirically confirmed by the next experiment.

### 4.4. Test Case: Binary MNIST and CIFAR100 Tasks

In light of the analysis made in the previous subsection, we now explore an experimental setup where a prior model cannot encompass most of the information of the various tasks. To do so, we create a variety of binary tasks involving the various classes of the MNIST (CIFAR100) dataset, where a task corresponds to a random class versus another one. We create 90 (150) such tasks, where the meta-test set corresponds to all of the tasks involving either label 0 or label 1, chosen at random, leading to a total of 34 (50) meta-test tasks. Each training task contains 2000 (1200) examples from the train split of the MNIST (CIFAR100) original task, while each test task contains at most 2000 (200) examples from the test split of the original task. We consider the same benchmarks as previously. We report the test bounds and test errors on the novel tasks of various methods in Table 2 with their corresponding latent representation information in Table 3.

As expected, in such a setting, when it comes to the benchmarks, the posterior for each task is required to be truly different from the prior in order to perform well (as attested by the strawman's test error, being similar to a random guess). We present in Table 4 the penalty (KL) value measuring the distance between the prior and posterior for two benchmarks on both the pixel swap experiments and the binary MNIST variant. There is a significant gap between the KL value reported for the various pixel swap tasks and the binary MNIST task, which confirms our insight. Thus, the benchmarks methods generate uninformative generalization bounds, even though their test loss is competitive. On the other hand, most of our approaches achieve competitive empirical performances while also having informative generalization bounds, since the downstream predictors can be truly different from one another without impacting the quality of the bound.

The architectures reported in Table 3, along with the empirical performances in Table 2, confirm that 1) the encoder / the message compressor and the sample compressor correctly distill the particularities of the task at hand, and 2) that the reconstructor is able to utilize this representation to judiciously generate the downstream predictor.

Figure 7 depicts the test error and generalization bound for our PB SCH algorithm as a function of both the compression set size and the message size. We recall that Tables 1 and 2 report the performances of the models obtaining the best validation error. Figure 7 helps to grasp the inner workings of our proposed approach: using a larger message seems better-suited for minimizing the test error, but a trade-off between the compression set size and the message size is required to obtain the best bounds. Interestingly, when the message size $|\boldsymbol{\mu}|$ is restricted to be small, we clearly see the benefit of using a compression set ($c > 0$).

Table 1: Comparison of different meta-learning methods on the MNIST-pixels-swap task. The 95% confidence interval is reported for generalization bound and test error, computed over 20 test tasks. The best (smallest) result in each column is **bolded**.

| Algorithm | 100 Pixels swap | | 200 Pixels swap | | 300 Pixels swap | |
|---|---|---|---|---|---|---|
| | Bound ($\downarrow$) | Test error ($\downarrow$) | Bound ($\downarrow$) | Test error ($\downarrow$) | Bound ($\downarrow$) | Test error ($\downarrow$) |
| (Pentina & Lampert, 2014) | $0.190 \pm 0.022$ | $0.019 \pm 0.001$ | $0.240 \pm 0.030$ | $0.026 \pm 0.002$ | $0.334 \pm 0.036$ | $0.038 \pm 0.003$ |
| (Amit & Meir, 2018) | $0.138 \pm 0.024$ | $0.016 \pm 0.001$ | $0.161 \pm 0.002$ | $0.020 \pm 0.001$ | $0.329 \pm 0.081$ | $0.040 \pm 0.681$ |
| (Guan & Lu, 2022) - kl | $0.119 \pm 0.024$ | $0.017 \pm 0.001$ | $0.189 \pm 0.027$ | $0.026 \pm 0.001$ | $0.359 \pm 0.042$ | $0.030 \pm 0.002$ |
| (Guan & Lu, 2022) - Catoni | $0.093 \pm 0.027$ | $\mathbf{0.015} \pm 0.001$ | $0.128 \pm 0.025$ | $\mathbf{0.019} \pm 0.001$ | $0.210 \pm 0.035$ | $\mathbf{0.024} \pm 0.001$ |
| (Zakerinia et al., 2024) | $0.053 \pm 0.020$ | $0.019 \pm 0.1346$ | $0.108 \pm 0.037$ | $0.026 \pm 0.263$ | $\mathbf{0.149} \pm 0.039$ | $0.035 \pm 0.547$ |
| PBH | $0.068^{*}\pm 0.007$ | $0.027 \pm 0.005$ | $0.112^{*}\pm 0.021$ | $0.076 \pm 0.018$ | $0.219^{*}\pm 0.031$ | $0.186 \pm 0.060$ |
| SCH$_{-}$ | $0.067 \pm 0.007$ | $0.029 \pm 0.007$ | $0.129 \pm 0.023$ | $0.084 \pm 0.017$ | $0.193 \pm 0.038$ | $0.162 \pm 0.043$ |
| SCH$_{+}$ | $\mathbf{0.035} \pm 0.012$ | $0.024 \pm 0.005$ | $\mathbf{0.091} \pm 0.022$ | $0.075 \pm 0.019$ | $0.177 \pm 0.032$ | $0.153 \pm 0.028$ |
| PB SCH | $0.068^{*}\pm 0.007$ | $0.027 \pm 0.005$ | $0.112^{*}\pm 0.021$ | $0.076 \pm 0.018$ | $0.219^{*}\pm 0.031$ | $0.186 \pm 0.060$ |
| Opaque encoder | $0.043 \pm 0.003$ | $0.037 \pm 0.006$ | $0.092 \pm 0.019$ | $0.087 \pm 0.018$ | $0.173 \pm 0.030$ | $0.159 \pm 0.031$ |

$^{*}$Bound on average over the decoder output.

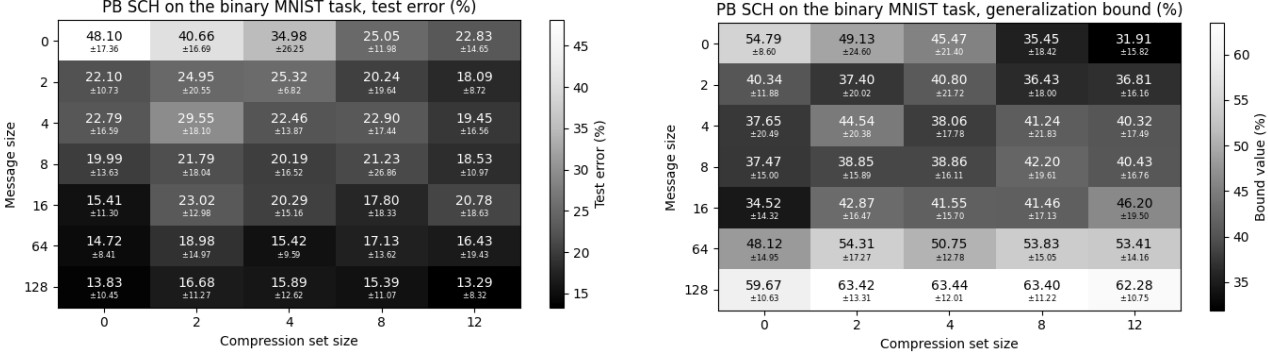

Figure 7: Test error and generalization bound for the PB SCH algorithm as a function of both the compression set size and the message size on binary MNIST tasks. The 95% confidence interval is reported for both the generalization bound and the test error, computed over 34 test tasks.

Table 2: Comparison of different meta-learning methods on the MNIST and CIFAR100 binary tasks. The 95% confidence interval is reported for generalization bound and test error, computed over test tasks. The best (smallest) result in each column is **bolded**.

| Algorithm | MNIST | | CIFAR100 | |
|---|---|---|---|---|
| | Bound ($\downarrow$) | Test error ($\downarrow$) | Bound ($\downarrow$) | Test error ($\downarrow$) |
| (Pentina & Lampert, 2014) | $0.767 \pm 0.001$ | $0.369 \pm 0.223$ | $0.801 \pm 0.001$ | $0.490 \pm 0.070$ |
| (Amit & Meir, 2018) | $1372 \pm 23.36$ | $0.351 \pm 0.212$ | $950.9 \pm 343.1$ | $0.284 \pm 0.120$ |
| (Guan & Lu, 2022) - kl | $0.754 \pm 0.003$ | $0.366 \pm 0.221$ | $0.802 \pm 0.001$ | $0.489 \pm 0.073$ |
| (Guan & Lu, 2022) - Cat. | $1.132 \pm 0.021$ | $0.351 \pm 0.212$ | $1.577 \pm 0.567$ | $0.282 \pm 0.122$ |
| (Rezazadeh, 2022) | $11.43 \pm 0.005$ | $0.366 \pm 0.221$ | $10.91 \pm 0.368$ | $0.334 \pm 0.139$ |
| (Zakerinia et al., 2024) | $0.684 \pm 0.021$ | $0.351 \pm 0.212$ | $0.953 \pm 0.315$ | $\mathbf{0.281} \pm 0.125$ |
| PBH | $0.597^{*}\pm 0.107$ | $\mathbf{0.150} \pm 0.114$ | $0.974^{*}\pm 0.022$ | $0.295 \pm 0.103$ |
| SCH$_{-}$ | $0.352 \pm 0.187$ | $0.278 \pm 0.076$ | $\mathbf{0.600} \pm 0.143$ | $0.374 \pm 0.118$ |
| SCH$_{+}$ | $\mathbf{0.280} \pm 0.148$ | $0.155 \pm 0.109$ | $0.745 \pm 0.101$ | $0.305 \pm 0.142$ |
| PB SCH | $0.597^{*}\pm 0.107$ | $\mathbf{0.150} \pm 0.114$ | $0.974^{*}\pm 0.022$ | $0.295 \pm 0.103$ |
| Opaque encoder | $0.533 \pm 0.104$ | $0.497 \pm 0.134$ | $0.544 \pm 0.112$ | $0.506 \pm 0.101$ |

$^{*}$Bound on average over the decoder output.

Table 3: Selected architecture for the sample compression hypernetwork algorithm (we recall that $c$ corresponds to the compression set size, while $|\boldsymbol{\mu}|$, $b$ corresponds to the message size); hyperparameter choices can be found in Appendix H.

| SC Hypernetwork | MNIST | | CIFAR100 | |
|---|---|---|---|---|
| | $c$ | $|\boldsymbol{\mu}|$, $b$ | $c$ | $|\boldsymbol{\mu}|$, $b$ |
| PBH | N/A | 128 | N/A | 128 |
| SCH$_{-}$ | 8 | N/A | 4 | N/A |
| SCH$_{+}$ | 1 | 64 | 1 | 128 |
| PB SCH | 0 | 128 | 0 | 128 |

See Table 5 in Appendix H for results involving the bound from Theorem 2.3 and Theorem 2.5. See also in Appendix H a decomposition of the various terms involved in the composition of our bounds and the benchmarks'.

Table 4: KL value of two benchmarks on the pixel swap tasks and the binary MNIST task. The 95% confidence interval is reported for generalization bound and test error, computed over 20 test tasks.

| Algorithm | 100 Pixels swap | 200 Pixels swap | 300 Pixels swap | binary MNIST |
|---|---|---|---|---|
| (Pentina & Lampert, 2014) | $3.833 \pm 0.444$ | $5.760 \pm 0.720$ | $9.604 \pm 1.035$ | $14.02 \pm 0.018$ |
| (Zakerinia et al., 2024) | $63.91 \pm 24.12$ | $159.9 \pm 54.78$ | $223.9 \pm 58.60$ | $661.2 \pm 20.32$ |

## 5. Conclusion

We developed a new paradigm for deriving generalization bounds in meta-learning by leveraging the PAC-Bayesian framework or the Sample Compression theory. We also present a new generalization bound that permits the coupling of both paradigms. We develop meta-learning hypernetworks based on these results. We show that many PAC-Bayes approaches do not scale when the various tasks in an environment have important discrepancies while our approaches still yield low losses and tight generalization bounds.

The approaches we presented could be enhanced by having the model dynamically select the compression set size and the latent representation (or message) size or using a larger architecture for the reconstruction function, inspired from Mother-Net (Mueller et al., 2024). Finally, since the bound values are not impacted by the complexity of the decoder or the downstream predictor, our approach could be used to get tight generalization bounds for very large models of the multi-billion parameter scale, assuming that the parameters varying across tasks admit a compact representation (e.g., a LoRA adapter).

## Impact Statement

This paper presents work whose goal is to advance the field of Machine Learning. Since we work for the better certification of machine learning algorithms, we do not see any potentially harmful societal consequences of our work.

## Acknowledgements

This research is supported by the NSERC/Intact Financial Corporation Industrial Research Chair in Machine Learning for Insurance. Pascal Germain is supported by the Canada CIFAR AI Chair Program, and the NSERC Discovery grant RGPIN-2020-07223. Mathieu Bazinet is supported by a FRQNT B2X scholarship (343192, doi: https://doi.org/10.69777/343192).

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

## A. Corollaries of Theorem 2.3

For completeness, we present the corollaries of Theorem 2.3 derived by Bazinet et al. (2025).

**Corollary A.1** (Corollary 4 of Bazinet et al. (2025)). *In the setting of Theorem 2.3, with*

$$\Delta_C(q,p) = -\ln\big(1 - (1 - e^{-C})p\big) - Cq \quad \text{(where } C > 0\text{)},$$

*with probability at least $1 - \delta$ over the draw of $S \sim \mathcal{D}^m$, we have*

$$\forall \mathbf{j} \in J, \omega \in M(|\mathbf{j}|) \; : $$
$$\mathcal{L}_D(\mathcal{R}(S_{\mathbf{j}}, \omega)) \leq \frac{1}{1 - e^{-C}}\left[1 - \exp\left(-C\widehat{\mathcal{L}}_{S_{\bar{\mathbf{j}}}}(\mathcal{R}(S_{\mathbf{j}}, \omega)) + \frac{\ln\big(P_J(\mathbf{j}) \cdot P_{M(|\mathbf{j}|)}(\omega) \cdot \delta\big)}{m - |\mathbf{j}|}\right)\right].$$

**Corollary A.2** (Corollary 6 of Bazinet et al. (2025)). *In the setting of Theorem 2.3, with*

$$\Delta(q,p) = \mathrm{kl}(q,p) = q \cdot \ln\frac{q}{p} + (1 - q) \cdot \ln\frac{1 - q}{1 - p},$$

*with probability at least $1 - \delta$ over the draw of $S \sim \mathcal{D}^m$, we have*

$$\forall \mathbf{j} \in J, \omega \in M(|\mathbf{j}|) \; : \; \mathrm{kl}\Big(\widehat{\mathcal{L}}_{S_{\bar{\mathbf{j}}}}(\mathcal{R}(S_{\mathbf{j}}, \omega)), \mathcal{L}_D(\mathcal{R}(S_{\mathbf{j}}, \omega))\Big) \leq \frac{1}{m - |\mathbf{j}|}\left[\ln\left(\frac{2\sqrt{m - |\mathbf{j}|}}{P_J(\mathbf{j}) \cdot P_{M(|\mathbf{j}|)}(\omega) \cdot \delta}\right)\right].$$

**Corollary A.3** (Corollary 7 of Bazinet et al. (2025)). *In the setting of Theorem 2.3, for any $\lambda > 0$, with $\Delta(q,p) = \lambda(p - q)$, with a $\varsigma^2$-sub-Gaussian loss function $\ell : \mathcal{Y} \times \mathcal{Y} \to \mathbb{R}$, with probability at least $1 - \delta$ over the draw of $S \sim \mathcal{D}^m$, we have*

$$\forall \mathbf{j} \in J, \omega \in M(|\mathbf{j}|) \; : \; \mathcal{L}_D(\mathcal{R}(S_{\mathbf{j}}, \omega)) \leq \widehat{\mathcal{L}}_{S_{\bar{\mathbf{j}}}}(\mathcal{R}(S_{\mathbf{j}}, \omega)) + \frac{\lambda\varsigma^2}{2} - \frac{\ln\big(P_J(\mathbf{j}) \cdot P_{M(|\mathbf{j}|)}(\omega) \cdot \delta\big)}{\lambda(m - |\mathbf{j}|)}.$$

## B. General PAC-Bayes Sample Compression Theorems

We derive two new PAC-Bayes Sample compression theorems for real-valued losses, both for the loss on the complement set and on the train set. Theorem B.1 extends the setting of Laviolette & Marchand (2005); **?**); Thiemann et al. (2017), whilst Theorem B.2 extends the setting of Germain et al. (2011; 2015).

### B.1. First General PAC-Bayes Sample Compression Theorem

We first present a new general PAC-Bayes Sample Compression theorem, from which we will derive a sample compression bound for continuous messages. Interestingly, this theorem is directly at the intersection between PAC-Bayes theory and sample compression theory. Indeed, if we restrict the model to have no compression set, this bound reduces to Theorem 2.1. Moreover, if we restrict the model to a discrete family $\Omega$ and to Dirac measures as posteriors on $J$ and $\Omega$, this bound is almost exactly reduced to Theorem 2.3, albeit being possibly slightly less tight, as the complexity term denominator $m - |\mathbf{j}|$ of the latter is replaced by the worst case: $m - \max_{\mathbf{j} \in J} |\mathbf{j}|$.

**Theorem B.1** (PAC-Bayes Sample Compression). *For any distribution $\mathcal{D}$ over $\mathcal{X} \times \mathcal{Y}$, for any reconstruction function $\mathcal{R}$, for any set $J \subseteq \mathcal{P}(\mathbf{m})$, for any set of messages $\Omega$, for any data-independent prior distribution $P$ over $J \times \Omega$, for any loss $\ell : \mathcal{Y} \times \mathcal{Y} \to [0, 1]$, for any convex function $\Delta : [0, 1] \times [0, 1] \to \mathbb{R}$ and for any $\delta \in (0, 1]$, with probability at least $1 - \delta$ over the draw of $S \sim \mathcal{D}^m$, we have*

$$\forall Q \text{ over } J \times \Omega \; :$$
$$\Delta\left(\mathop{\mathbb{E}}_{(\mathbf{j},\omega)\sim Q} \widehat{\mathcal{L}}_{S_{\bar{\mathbf{j}}}}(\mathcal{R}(S_{\mathbf{j}}, \omega)), \mathop{\mathbb{E}}_{(\mathbf{j},\omega)\sim Q} \mathcal{L}_{\mathcal{D}}(\mathcal{R}(S_{\mathbf{j}}, \omega))\right) \leq \frac{1}{m - \max_{\mathbf{j} \in J} |\mathbf{j}|}\left[\mathrm{KL}(Q||P) + \ln\left(\frac{\mathcal{A}_\Delta(m)}{\delta}\right)\right],$$

*with*

$$\mathcal{A}_\Delta(m) = \mathop{\mathbb{E}}_{(\mathbf{j},\omega)\sim P} \mathop{\mathbb{E}}_{T_{\mathbf{j}}\sim\mathcal{D}^{|\mathbf{j}|}} \mathop{\mathbb{E}}_{T_{\bar{\mathbf{j}}}\sim\mathcal{D}^{m-|\mathbf{j}|}} e^{(m-|\mathbf{j}|)\Delta\left(\widehat{\mathcal{L}}_{T_{\bar{\mathbf{j}}}}(\mathcal{R}(T_{\mathbf{j}},\omega)), \mathcal{L}_{\mathcal{D}}(\mathcal{R}(T_{\mathbf{j}},\omega))\right)}.$$

*Proof.* With $\eta > 0$, our goal is to bound the following expression

$$\eta\Delta\left(\mathop{\mathbb{E}}_{(\mathbf{j},\omega)\sim Q} \widehat{\mathcal{L}}_{S_{\bar{\mathbf{j}}}}\left(\mathcal{R}(S_{\mathbf{j}},\omega)\right), \mathop{\mathbb{E}}_{(\mathbf{j},\omega)\sim Q} \mathcal{L}_{\mathcal{D}}\left(\mathcal{R}(S_{\mathbf{j}},\omega)\right)\right).$$

We follow the proof of the General PAC-Bayes bound for real-valued losses of Germain et al. (2015). We first apply Jensen's inequality and then use the change of measure inequality to obtain the following result.

$$\forall Q \text{ over } J \times \Omega: \quad \eta\Delta\left(\mathop{\mathbb{E}}_{(\mathbf{j},\omega)\sim Q} \widehat{\mathcal{L}}_{S_{\bar{\mathbf{j}}}}\left(\mathcal{R}(S_{\mathbf{j}},\omega)\right), \mathop{\mathbb{E}}_{(\mathbf{j},\omega)\sim Q} \mathcal{L}_{\mathcal{D}}\left(\mathcal{R}(S_{\mathbf{j}},\omega)\right)\right)$$

$$\leq \mathop{\mathbb{E}}_{(\mathbf{j},\omega)\sim Q} \eta\Delta\left(\widehat{\mathcal{L}}_{S_{\bar{\mathbf{j}}}}\left(\mathcal{R}(S_{\mathbf{j}},\omega)\right), \mathcal{L}_{\mathcal{D}}\left(\mathcal{R}(S_{\mathbf{j}},\omega)\right)\right) \qquad \text{(Jensen's Inequality)}$$

$$\leq \mathrm{KL}(Q\|P) + \ln\left(\mathop{\mathbb{E}}_{(\mathbf{j},\omega)\sim P} e^{\eta\Delta\left(\widehat{\mathcal{L}}_{S_{\bar{\mathbf{j}}}}\left(\mathcal{R}(S_{\mathbf{j}},\omega)\right), \mathcal{L}_{\mathcal{D}}\left(\mathcal{R}(S_{\mathbf{j}},\omega)\right)\right)}\right). \qquad \text{(Change of measure)}$$

Using Markov's inequality, we know that with probability at least $1 - \delta$ over the sampling of $S \sim \mathcal{D}^n$, we have

$$\forall Q \text{ over } J \times \Omega:$$

$$\eta\Delta\left(\mathop{\mathbb{E}}_{(\mathbf{j},\omega)\sim Q} \widehat{\mathcal{L}}_{S_{\bar{\mathbf{j}}}}\left(\mathcal{R}(S_{\mathbf{j}},\omega)\right), \mathop{\mathbb{E}}_{(\mathbf{j},\omega)\sim Q} \mathcal{L}_{\mathcal{D}}\left(\mathcal{R}(S_{\mathbf{j}},\omega)\right)\right) \leq \mathrm{KL}(Q\|P) + \ln\left(\frac{1}{\delta} \mathop{\mathbb{E}}_{T\sim\mathcal{D}^m} \mathop{\mathbb{E}}_{(\mathbf{j},\omega)\sim P} e^{\eta\Delta\left(\widehat{\mathcal{L}}_{T_{\bar{\mathbf{j}}}}\left(\mathcal{R}(T_{\mathbf{j}},\omega)\right), \mathcal{L}_{\mathcal{D}}\left(\mathcal{R}(T_{\mathbf{j}},\omega)\right)\right)}\right).$$

By choosing to define $Q$ and $P$ on $J \times \Omega$ instead of on the hypothesis class, the prior $P$ is independent of the dataset. We can then swap the expectations to finish the proof.

We use the independence of the prior to the dataset and the *i.i.d.* assumption to separate $T_{\mathbf{j}}$ and $T_{\bar{\mathbf{j}}} = T \setminus T_{\mathbf{j}}$ to obtain the following results.

$$\mathop{\mathbb{E}}_{T\sim\mathcal{D}^m} \mathop{\mathbb{E}}_{(\mathbf{j},\omega)\sim P} e^{\eta\Delta\left(\widehat{\mathcal{L}}_{T_{\bar{\mathbf{j}}}}\left(\mathcal{R}(T_{\mathbf{j}},\omega)\right), \mathcal{L}_{\mathcal{D}}\left(\mathcal{R}(T_{\mathbf{j}},\omega)\right)\right)}$$

$$= \mathop{\mathbb{E}}_{(\mathbf{j},\omega)\sim P} \mathop{\mathbb{E}}_{T\sim\mathcal{D}^m} e^{\eta\Delta\left(\widehat{\mathcal{L}}_{T_{\bar{\mathbf{j}}}}\left(\mathcal{R}(T_{\mathbf{j}},\omega)\right), \mathcal{L}_{\mathcal{D}}\left(\mathcal{R}(T_{\mathbf{j}},\omega)\right)\right)} \qquad \text{(Independence of the prior)}$$

$$= \mathop{\mathbb{E}}_{(\mathbf{j},\omega)\sim P} \mathop{\mathbb{E}}_{T_{\mathbf{j}}\sim\mathcal{D}^{|\mathbf{j}|}} \mathop{\mathbb{E}}_{T_{\bar{\mathbf{j}}}\sim\mathcal{D}^{m-|\mathbf{j}|}} e^{\eta\Delta\left(\widehat{\mathcal{L}}_{T_{\bar{\mathbf{j}}}}\left(\mathcal{R}(T_{\mathbf{j}},\omega)\right), \mathcal{L}_{\mathcal{D}}\left(\mathcal{R}(T_{\mathbf{j}},\omega)\right)\right)}. \qquad \text{(\textit{i.i.d.} assumption)}$$

For all $\mathbf{j} \in J$, we need to bound the moment generating function

$$\mathop{\mathbb{E}}_{T_{\mathbf{j}}\sim\mathcal{D}^{|\mathbf{j}|}} \mathop{\mathbb{E}}_{T_{\bar{\mathbf{j}}}\sim\mathcal{D}^{m-|\mathbf{j}|}} e^{\eta\Delta\left(\widehat{\mathcal{L}}_{T_{\bar{\mathbf{j}}}}\left(\mathcal{R}(T_{\mathbf{j}},\omega)\right), \mathcal{L}_{\mathcal{D}}\left(\mathcal{R}(T_{\mathbf{j}},\omega)\right)\right)}.$$

To bound the moment generating function using the usual PAC-Bayes techniques, we need $\eta \leq m - |\mathbf{j}|$ for all $\mathbf{j} \in J$. Thus, the largest value of $\eta$ (which gives the tightest bound) that can be used is $\eta = m - \max_{\mathbf{j}\in J} |\mathbf{j}|$.

As the exponential is monotonically increasing and $m - \max_{\mathbf{j}\in J} |\mathbf{j}| \leq m - |\mathbf{j}| \, \forall \mathbf{j}$, we have

$$\mathop{\mathbb{E}}_{T_{\mathbf{j}}\sim\mathcal{D}^{|\mathbf{j}|}} \mathop{\mathbb{E}}_{T_{\bar{\mathbf{j}}}\sim\mathcal{D}^{m-|\mathbf{j}|}} e^{\eta\Delta\left(\widehat{\mathcal{L}}_{T_{\bar{\mathbf{j}}}}\left(\mathcal{R}(T_{\mathbf{j}},\omega)\right), \mathcal{L}_{\mathcal{D}}\left(\mathcal{R}(T_{\mathbf{j}},\omega)\right)\right)} \leq \mathop{\mathbb{E}}_{T_{\mathbf{j}}\sim\mathcal{D}^{|\mathbf{j}|}} \mathop{\mathbb{E}}_{T_{\bar{\mathbf{j}}}\sim\mathcal{D}^{m-|\mathbf{j}|}} e^{(m-|\mathbf{j}|)\Delta\left(\widehat{\mathcal{L}}_{T_{\bar{\mathbf{j}}}}\left(\mathcal{R}(T_{\mathbf{j}},\omega)\right), \mathcal{L}_{\mathcal{D}}\left(\mathcal{R}(T_{\mathbf{j}},\omega)\right)\right)} =: \mathcal{A}_{\Delta}(m).$$

$\square$

## B.2. Second General PAC-Bayes Sample Compression Theorem

We now extend the setting of Germain et al. (2015), by making the bound rely on $\widehat{\mathcal{L}}_S(\cdot)$ instead of $\widehat{\mathcal{L}}_{S_{\bar{\mathbf{j}}}}(\cdot)$.

**Theorem B.2** (PAC-Bayes Sample Compression). *For any distribution $\mathcal{D}$ over $\mathcal{X} \times \mathcal{Y}$, for any reconstruction function $\mathcal{R}$, for any set $J \subseteq \mathcal{P}(\mathbf{m})$, for any set of messages $\Omega$, for any data-independent prior distribution $P$ over $J \times \Omega$, for any loss $\ell : \mathcal{Y} \times \mathcal{Y} \to [0, 1]$, for any convex function $\Delta : [0, 1] \times [0, 1] \to \mathbb{R}$, with $B = \sup_{q,p \in [0,1]} \Delta(q, p)$, and for any $\delta \in (0, 1]$, with probability at least $1 - \delta$ over the draw of $S \sim \mathcal{D}^m$, we have*

$$\forall Q \text{ over } J \times \Omega :$$

$$\Delta\left(\mathop{\mathbb{E}}_{(\mathbf{j},\omega) \sim Q} \widehat{\mathcal{L}}_S\left(\mathcal{R}(S_{\mathbf{j}}, \omega)\right), \mathop{\mathbb{E}}_{(\mathbf{j},\omega) \sim Q} \mathcal{L}_{\mathcal{D}}\left(\mathcal{R}(S_{\mathbf{j}}, \omega)\right)\right) \le \frac{1}{m - \max_{\mathbf{j} \in J} |\mathbf{j}|} \left[ \mathrm{KL}(Q\|P) + \ln\left(\frac{\mathcal{A}'_{\Delta}(m)}{\delta}\right) \right],$$

*with*

$$\mathcal{A}'_{\Delta}(m) = \mathop{\mathbb{E}}_{(\mathbf{j},\omega) \sim P} e^{B \frac{(m - \max_{\mathbf{j} \in J} |\mathbf{j}|) |\mathbf{j}|}{m}} \mathop{\mathbb{E}}_{T_{\mathbf{j}} \sim \mathcal{D}^{|\mathbf{j}|}} \mathop{\mathbb{E}}_{T_{\bar{\mathbf{j}}} \sim \mathcal{D}^{m-|\mathbf{j}|}} e^{(m-|\mathbf{j}|)\Delta\left(\widehat{\mathcal{L}}_{T_{\bar{\mathbf{j}}}}(\mathcal{R}(T_{\mathbf{j}}, \omega)), \mathcal{L}_{\mathcal{D}}(\mathcal{R}(T_{\mathbf{j}}, \omega))\right)}.$$

*Proof.* We start from the end of the proof of Theorem B.1, after using the *i.i.d.* assumption. As the loss of the hypothesis is computed on the data used to define $\mathcal{R}(T_{\mathbf{j}}, \omega)$, this term cannot be bounded straightforwardly as before. To tackle this problem, we assume that $\Delta$ is bounded. We want to remove the need to compute the loss on examples $T_{\mathbf{j}} = T \setminus T_{\bar{\mathbf{j}}}$ in the following term.

$$\mathop{\mathbb{E}}_{(\mathbf{j},\omega) \sim P} \mathop{\mathbb{E}}_{T_{\mathbf{j}} \sim \mathcal{D}^{|\mathbf{j}|}} \mathop{\mathbb{E}}_{T_{\bar{\mathbf{j}}} \sim \mathcal{D}^{m-|\mathbf{j}|}} e^{\eta\Delta\left(\widehat{\mathcal{L}}_T(\mathcal{R}(T_{\mathbf{j}}, \omega)), \mathcal{L}_{\mathcal{D}}(\mathcal{R}(T_{\mathbf{j}}, \omega))\right)} \quad \text{(with } \eta > 0\text{)}.$$

Following the work of Germain et al. (2015), we separate the loss :

$$\begin{aligned}
\widehat{\mathcal{L}}_T(\mathcal{R}(T_{\mathbf{j}}, \omega)) &= \frac{1}{m} \sum_{i=1}^m \ell(\mathcal{R}(T_{\mathbf{j}}, \omega)(\boldsymbol{x}_i), y_i) \\
&= \frac{1}{m} \left[ \sum_{(\boldsymbol{x}_i, y_i) \in T_{\mathbf{j}}} \ell(\mathcal{R}(T_{\mathbf{j}}, \omega)(\boldsymbol{x}_i), y_i) + \sum_{(\boldsymbol{x}_i, y_i) \in T_{\bar{\mathbf{j}}}} \ell(\mathcal{R}(T_{\mathbf{j}}, \omega)(\boldsymbol{x}_i), y_i) \right] \\
&= \frac{1}{m} \left[ \frac{|\mathbf{j}|}{|\mathbf{j}|} \sum_{(\boldsymbol{x}_i, y_i) \in T_{\mathbf{j}}} \ell(\mathcal{R}(T_{\mathbf{j}}, \omega)(\boldsymbol{x}_i), y_i) + \frac{m - |\mathbf{j}|}{m - |\mathbf{j}|} \sum_{(\boldsymbol{x}_i, y_i) \in T_{\bar{\mathbf{j}}}} \ell(\mathcal{R}(T_{\mathbf{j}}, \omega)(\boldsymbol{x}_i), y_i) \right] \\
&= \frac{1}{m} \left[ |\mathbf{j}| \widehat{\mathcal{L}}_{T_{\mathbf{j}}}(\mathcal{R}(T_{\mathbf{j}}, \omega)) + (m - |\mathbf{j}|) \widehat{\mathcal{L}}_{T_{\bar{\mathbf{j}}}}(\mathcal{R}(T_{\mathbf{j}}, \omega)) \right].
\end{aligned}$$

With this new expression, we have

$$\begin{aligned}
&\mathop{\mathbb{E}}_{(\mathbf{j},\omega) \sim P} \mathop{\mathbb{E}}_{T_{\mathbf{j}} \sim \mathcal{D}^{|\mathbf{j}|}} \mathop{\mathbb{E}}_{T_{\bar{\mathbf{j}}} \sim \mathcal{D}^{m-|\mathbf{j}|}} e^{\eta\Delta\left(\widehat{\mathcal{L}}_T(\mathcal{R}(T_{\mathbf{j}}, \omega)), \mathcal{L}_{\mathcal{D}}(\mathcal{R}(T_{\mathbf{j}}, \omega))\right)} \\
&= \mathop{\mathbb{E}}_{(\mathbf{j},\omega) \sim P} \mathop{\mathbb{E}}_{T_{\mathbf{j}} \sim \mathcal{D}^{|\mathbf{j}|}} \mathop{\mathbb{E}}_{T_{\bar{\mathbf{j}}} \sim \mathcal{D}^{m-|\mathbf{j}|}} e^{\eta\Delta\left(\frac{|\mathbf{j}|}{m} \widehat{\mathcal{L}}_{T_{\mathbf{j}}}(\mathcal{R}(T_{\mathbf{j}}, \omega)) + \frac{m-|\mathbf{j}|}{m} \widehat{\mathcal{L}}_{T_{\bar{\mathbf{j}}}}(\mathcal{R}(T_{\mathbf{j}}, \omega)), \mathcal{L}_{\mathcal{D}}(\mathcal{R}(T_{\mathbf{j}}, \omega))\right)} \\
&\le \mathop{\mathbb{E}}_{(\mathbf{j},\omega) \sim P} \mathop{\mathbb{E}}_{T_{\mathbf{j}} \sim \mathcal{D}^{|\mathbf{j}|}} \mathop{\mathbb{E}}_{T_{\bar{\mathbf{j}}} \sim \mathcal{D}^{m-|\mathbf{j}|}} e^{\frac{\eta|\mathbf{j}|}{m} \Delta\left(\widehat{\mathcal{L}}_{T_{\mathbf{j}}}(\mathcal{R}(T_{\mathbf{j}}, \omega)), \mathcal{L}_{\mathcal{D}}(\mathcal{R}(T_{\mathbf{j}}, \omega))\right) + \frac{\eta(m-|\mathbf{j}|)}{m} \Delta\left(\widehat{\mathcal{L}}_{T_{\bar{\mathbf{j}}}}(\mathcal{R}(T_{\mathbf{j}}, \omega)), \mathcal{L}_{\mathcal{D}}(\mathcal{R}(T_{\mathbf{j}}, \omega))\right)} \quad \text{(Jensen inequality)} \\
&\le \mathop{\mathbb{E}}_{(\mathbf{j},\omega) \sim P} \mathop{\mathbb{E}}_{T_{\mathbf{j}} \sim \mathcal{D}^{|\mathbf{j}|}} \mathop{\mathbb{E}}_{T_{\bar{\mathbf{j}}} \sim \mathcal{D}^{m-|\mathbf{j}|}} e^{\frac{\eta|\mathbf{j}|}{m} \sup_{q,p \in [0,1]} \Delta(q,p) + \frac{\eta(m-|\mathbf{j}|)}{m} \Delta\left(\widehat{\mathcal{L}}_{T_{\bar{\mathbf{j}}}}(\mathcal{R}(T_{\mathbf{j}}, \omega)), \mathcal{L}_{\mathcal{D}}(\mathcal{R}(T_{\mathbf{j}}, \omega))\right)} \\
&= \mathop{\mathbb{E}}_{(\mathbf{j},\omega) \sim P} e^{\frac{\eta|\mathbf{j}|}{m} \sup_{q,p \in [0,1]} \Delta(q,p)} \mathop{\mathbb{E}}_{T_{\mathbf{j}} \sim \mathcal{D}^{|\mathbf{j}|}} \mathop{\mathbb{E}}_{T_{\bar{\mathbf{j}}} \sim \mathcal{D}^{m-|\mathbf{j}|}} e^{\frac{\eta(m-|\mathbf{j}|)}{m} \Delta\left(\widehat{\mathcal{L}}_{T_{\bar{\mathbf{j}}}}(\mathcal{R}(T_{\mathbf{j}}, \omega)), \mathcal{L}_{\mathcal{D}}(\mathcal{R}(T_{\mathbf{j}}, \omega))\right)} \\
&= \mathop{\mathbb{E}}_{(\mathbf{j},\omega) \sim P} e^{B \frac{\eta|\mathbf{j}|}{m}} \mathop{\mathbb{E}}_{T_{\mathbf{j}} \sim \mathcal{D}^{|\mathbf{j}|}} \mathop{\mathbb{E}}_{T_{\bar{\mathbf{j}}} \sim \mathcal{D}^{m-|\mathbf{j}|}} e^{\frac{\eta(m-|\mathbf{j}|)}{m} \Delta\left(\widehat{\mathcal{L}}_{T_{\bar{\mathbf{j}}}}(\mathcal{R}(T_{\mathbf{j}}, \omega)), \mathcal{L}_{\mathcal{D}}(\mathcal{R}(T_{\mathbf{j}}, \omega))\right)} \quad (B := \sup_{q,p \in [0,1]} \Delta(q, p)) \\
&\le \mathop{\mathbb{E}}_{(\mathbf{j},\omega) \sim P} e^{B \frac{\eta|\mathbf{j}|}{m}} \mathop{\mathbb{E}}_{T_{\mathbf{j}} \sim \mathcal{D}^{|\mathbf{j}|}} \mathop{\mathbb{E}}_{T_{\bar{\mathbf{j}}} \sim \mathcal{D}^{m-|\mathbf{j}|}} e^{\eta\Delta\left(\widehat{\mathcal{L}}_{T_{\bar{\mathbf{j}}}}(\mathcal{R}(T_{\mathbf{j}}, \omega)), \mathcal{L}_{\mathcal{D}}(\mathcal{R}(T_{\mathbf{j}}, \omega))\right)}.
\end{aligned}$$

Similarly to Theorem B.1, we choose $\eta = m - \max_{\mathbf{j} \in J} |\mathbf{j}|$ and we have

$$
\mathop{\mathbb{E}}_{(\mathbf{j},\omega) \sim P} e^{B \frac{\eta |\mathbf{j}|}{m}} \mathop{\mathbb{E}}_{T_{\mathbf{j}} \sim \mathcal{D}^{|\mathbf{j}|}} \mathop{\mathbb{E}}_{T_{\bar{\mathbf{j}}} \sim \mathcal{D}^{m-|\mathbf{j}|}} e^{\eta \Delta \left( \widehat{\mathcal{L}}_{T_{\bar{\mathbf{j}}}}(\mathcal{R}(T_{\mathbf{j}},\omega)), \mathcal{L}_{\mathcal{D}}(\mathcal{R}(T_{\mathbf{j}},\omega)) \right)}
$$

$$
\leq \mathop{\mathbb{E}}_{(\mathbf{j},\omega) \sim P} e^{B \frac{(m - \max_{\mathbf{j} \in J} |\mathbf{j}|) |\mathbf{j}|}{m}} \mathop{\mathbb{E}}_{T_{\mathbf{j}} \sim \mathcal{D}^{|\mathbf{j}|}} \mathop{\mathbb{E}}_{T_{\bar{\mathbf{j}}} \sim \mathcal{D}^{m-|\mathbf{j}|}} e^{(m-|\mathbf{j}|) \Delta \left( \widehat{\mathcal{L}}_{T_{\bar{\mathbf{j}}}}(\mathcal{R}(T_{\mathbf{j}},\omega)), \mathcal{L}_{\mathcal{D}}(\mathcal{R}(T_{\mathbf{j}},\omega)) \right)} =: \mathcal{A}'_{\Delta}(m). \qquad \square
$$

The most common comparator functions, namely the quadratic loss $\Delta_2(q,p) = 2(q-p)^2$, Catoni's distance $\Delta_C(q,p) = -\ln\left(1 - (1 - e^{-C})p\right) - Cq$ and the linear distance $\Delta_\lambda(q,p) = \lambda(q-p)$ are all bounded, with the exception of the kl. Thus, we have $\sup_{q,p \in [0,1]} \Delta_2(q,p) = 2$, $\sup_{q,p \in [0,1]} \Delta_C(q,p) = C$ and $\sup_{q,p \in [a,b]} \Delta_\lambda(q,p) = \lambda(b-a)$.

## C. Corollaries of Theorem B.1 and Theorem B.2

With Theorem B.1 and $\Delta = \mathrm{kl}$, we recover a real-valued version of Theorem 4 of Laviolette & Marchand (2005).

**Corollary C.1.** *In the setting of Theorem B.1, with $\Delta(q,p) = \mathrm{kl}(q,p)$ and $S = S_{\bar{\mathbf{j}}}$, with probability at least $1 - \delta$ over the draw of $S \sim \mathcal{D}^m$, we have*

$$\forall Q \text{ over } J \times \Omega:$$

$$
\mathrm{kl}\left( \mathop{\mathbb{E}}_{(\mathbf{j},\omega) \sim Q} \widehat{\mathcal{L}}_{S_{\bar{\mathbf{j}}}}(\mathcal{R}(S_{\mathbf{j}},\omega)), \mathop{\mathbb{E}}_{(\mathbf{j},\omega) \sim Q} \mathcal{L}_{\mathcal{D}}(\mathcal{R}(S_{\mathbf{j}},\omega)) \right) \leq \frac{1}{m - \max_{\mathbf{j} \in J} |\mathbf{j}|} \left[ \mathrm{KL}(Q||P) + \ln\left( \frac{\mathbb{E}_{(\mathbf{j},\omega) \sim P} 2\sqrt{m - |\mathbf{j}|}}{\delta} \right) \right].
$$

We can relax this corollary to obtain a bound that has Theorem 6 of Thiemann et al. (2017) as a special case where we choose $J = \{\mathbf{j} \mid |\mathbf{j}| = r\}$ for $r > 0$.

**Corollary C.2.** *In the setting of Theorem B.1, with $S = S_{\bar{\mathbf{j}}}$, with probability at least $1 - \delta$ over the draw of $S \sim \mathcal{D}^m$, we have*

$$\forall Q \text{ over } J \times \Omega, \forall \lambda \in (0,2):$$

$$
\mathop{\mathbb{E}}_{(\mathbf{j},\omega) \sim Q} \widehat{\mathcal{L}}_{S_{\bar{\mathbf{j}}}}(\mathcal{R}(S_{\mathbf{j}},\omega)) \leq \frac{\mathbb{E}_{(\mathbf{j},\omega) \sim Q} \mathcal{L}_{\mathcal{D}}(\mathcal{R}(S_{\mathbf{j}},\omega))}{1 - \frac{\lambda}{2}} + \frac{\mathrm{KL}(Q||P) + \ln\left( \frac{1}{\delta} \mathbb{E}_{(\mathbf{j},\omega) \sim P} 2\sqrt{m - |\mathbf{j}|} \right)}{\lambda(1 - \frac{\lambda}{2})(m - \max_{\mathbf{j} \in J} |\mathbf{j}|)}.
$$

We can also recover a tighter bound than Theorem 39 of Germain et al. (2015). Instead of bounding the complement loss, they bound the training loss on the whole dataset, thus we can use Theorem B.2. Although this could theoretically lead to tighter bounds, later in the section we present a toy experiment that shows that in practice, it rarely is tighter.

**Corollary C.3.** *In the setting of Theorem B.2, with $\Delta(q,p) = 2(q-p)^2$, with probability at least $1 - \delta$ over the draw $S \sim \mathcal{D}^m$, we have*

$$\forall Q \text{ over } J \times \Omega:$$

$$
\mathop{\mathbb{E}}_{(\mathbf{j},\omega) \sim Q} \mathcal{L}_{\mathcal{D}}(\mathcal{R}(S_{\mathbf{j}},\omega)) \leq \mathop{\mathbb{E}}_{(\mathbf{j},\omega) \sim Q} \widehat{\mathcal{L}}_S(\mathcal{R}(S_{\mathbf{j}},\omega)) + \sqrt{\frac{\mathrm{KL}(Q||P) + \ln\left( \frac{1}{\delta} \mathbb{E}_{(\mathbf{j},\omega) \sim P} e^{\frac{2(m-|\mathbf{j}|)|\mathbf{j}|}{m}} 2\sqrt{m - |\mathbf{j}|} \right)}{2(m - \max_{\mathbf{j} \in J} |\mathbf{j}|)}}.
$$

Corollary C.3 expresses a tighter and more general bound than the results of Germain et al. (2015). Moreover, as explained below, the latter is not always valid.

**Identifying a small error in previous work of Germain et al. (2015).** Although the sample compression result of Germain et al. (2015) is stated for any compression sets of size at most $\lambda$, only the case for compression set sizes $\lambda$ were considered by the authors. Indeed, in the proof of Lemma 38, with $\lambda > 0$, when bounding $\mathcal{A}'_{\Delta}(m)$, they consider the term

$$
\mathop{\mathbb{E}}_{(\mathbf{j},\omega) \sim P} \mathop{\mathbb{E}}_{S_{\mathbf{j}} \sim \mathcal{D}^\lambda} \mathop{\mathbb{E}}_{S_{\bar{\mathbf{j}}} \sim \mathcal{D}^{m-\lambda}} e^{(m-\lambda)2\left( \widehat{\mathcal{L}}_S(\mathcal{R}(S_{\mathbf{j}},\omega)) - \mathcal{L}_{\mathcal{D}}(\mathcal{R}(S_{\mathbf{j}},\omega)) \right)^2}. \tag{8}
$$

By the definition $S_{\mathbf{j}} = \{(\boldsymbol{x}_j, y_j)\}_{j \in \mathbf{j}}$, we have that $|S_{\mathbf{j}}| = |\mathbf{j}|$. Thus, Equation (8) only makes sense when $\lambda = |\mathbf{j}|$. Moreover, later on, the decomposition of the loss

$$\widehat{\mathcal{L}}_S(\mathcal{R}(S_{\mathbf{j}}, \omega)) = \frac{1}{m}\left[\lambda \widehat{\mathcal{L}}_{S_{\mathbf{j}}}(\mathcal{R}(S_{\mathbf{j}}, \omega)) + (m - \lambda) \widehat{\mathcal{L}}_{S_{\bar{\mathbf{j}}}}(\mathcal{R}(S_{\mathbf{j}}, \omega))\right]$$

only makes sense when $|\mathbf{j}| = \lambda$. Indeed, we have

$$\begin{aligned}
\widehat{\mathcal{L}}_S(\mathcal{R}(S_{\mathbf{j}}, \omega)) &= \frac{1}{m}\sum_{i=1}^{m} \ell(\mathcal{R}(S_{\mathbf{j}}, \omega)(\boldsymbol{x}_i), y_i) \\
&= \frac{1}{m}\left[\sum_{(\boldsymbol{x}_i, y_i) \in S_{\mathbf{j}}} \ell(\mathcal{R}(S_{\mathbf{j}}, \omega)(\boldsymbol{x}_i), y_i) + \sum_{(\boldsymbol{x}_i, y_i) \in S_{\bar{\mathbf{j}}}} \ell(\mathcal{R}(S_{\mathbf{j}}, \omega)(\boldsymbol{x}_i), y_i)\right] \\
&= \frac{1}{m}\left[\frac{\lambda}{\lambda}\sum_{(\boldsymbol{x}_i, y_i) \in S_{\mathbf{j}}} \ell(\mathcal{R}(S_{\mathbf{j}}, \omega)(\boldsymbol{x}_i), y_i) + \frac{m - \lambda}{m - \lambda}\sum_{(\boldsymbol{x}_i, y_i) \in S_{\bar{\mathbf{j}}}} \ell(\mathcal{R}(S_{\mathbf{j}}, \omega)(\boldsymbol{x}_i), y_i)\right] \\
&= \frac{1}{m}\left[\lambda \widehat{\mathcal{L}}_{S_{\mathbf{j}}}(\mathcal{R}(S_{\mathbf{j}}, \omega)) + (m - \lambda) \widehat{\mathcal{L}}_{S_{\bar{\mathbf{j}}}}(\mathcal{R}(S_{\mathbf{j}}, \omega))\right].
\end{aligned}$$

If $\lambda \neq |\mathbf{j}|$, then $\widehat{\mathcal{L}}_{S_{\mathbf{j}}}(\mathcal{R}(S_{\mathbf{j}}, \omega)) \neq \frac{1}{\lambda}\sum_{(\boldsymbol{x}_i, y_i) \in S_{\mathbf{j}}} \ell(\mathcal{R}(S_{\mathbf{j}}, \omega)(\boldsymbol{x}_i), y_i)$ and $\widehat{\mathcal{L}}_{S_{\bar{\mathbf{j}}}}(\mathcal{R}(S_{\mathbf{j}}, \omega)) \neq \frac{1}{m - \lambda}\sum_{(\boldsymbol{x}_i, y_i) \in S_{\bar{\mathbf{j}}}} \ell(\mathcal{R}(S_{\mathbf{j}}, \omega)(\boldsymbol{x}_i), y_i)$.

The result was used correctly in Germain et al. (2015), as they only considered this setting. However, the statement of the theorem was slightly incorrect.

In this specific setting, we can easily show that the log term of Corollary C.3 is upper bounded by the log term in Theorem 39 of Germain et al. (2015):

$$\begin{aligned}
\frac{1}{\delta}e^{\frac{2(m - |\mathbf{j}|)|\mathbf{j}|}{m}} 2\sqrt{m - |\mathbf{j}|} &\leq \frac{1}{\delta}e^{2|\mathbf{j}|} 2\sqrt{m - |\mathbf{j}|} \\
&\leq \frac{1}{\delta}e^{4|\mathbf{j}|} 2\sqrt{m - |\mathbf{j}|}.
\end{aligned}$$

**Empirical comparison of the bounds.** We now compare the bound on $\widehat{\mathcal{L}}_S$ instead of $\widehat{\mathcal{L}}_{S_{\bar{\mathbf{j}}}}$. To do so, we compute the value of Corollary C.3 and a relaxed version of Corollary C.1 using Pinsker's inequality. We choose $m = 10000$, $\mathrm{KL}(Q||P) = 100$, $\delta = 0.01$, $\widehat{\mathcal{L}}_{S_{\mathbf{j}}}(h) = 0$ and $J$ such that $\max_{\mathbf{j} \in J} |\mathbf{j}| = 2000$. In this setting, we can investigate the relationship between the compression set size and the validation loss $\widehat{\mathcal{L}}_{S_{\bar{\mathbf{j}}}}(h)$. In Figure 8(a), we report the difference between Corollary C.3 and the relaxed version of Corollary C.1.

It is obvious that in this setting, the exponential term $\exp\left(\frac{2(m - |\mathbf{j}|)|\mathbf{j}|}{m}\right)$ of Corollary C.3 is very penalizing. We can see that the difference becomes smaller when the validation loss becomes larger. This can be explained by the fact that the loss $\widehat{\mathcal{L}}_S(h) = \frac{m - |\mathbf{j}|}{m}\widehat{\mathcal{L}}_{S_{\bar{\mathbf{j}}}}(h)$ decreases when $|\mathbf{j}|$ becomes larger. However, it doesn't come close to being advantageous. Indeed, it seems like Corollary C.3 only becomes advantageous when the KL is very large. In Figure 8(b), we chose $\mathrm{KL}(Q||P) = 10000$. We then observe that for $\widehat{\mathcal{L}}_{S_{\bar{\mathbf{j}}}}(h)$ larger than 0.6, Corollary C.3 is actually smaller. However, the advantage doesn't seem good enough to use this bound in any setting, as the advantage is only available in degenerate cases when the loss on the dataset is very high and the KL is also very large.

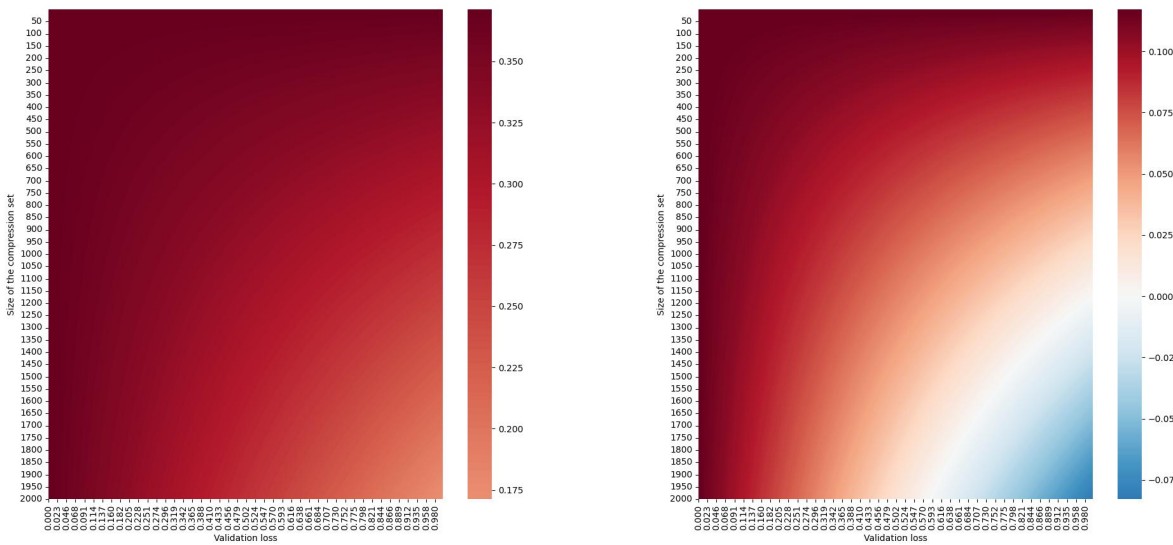

(a) The KL is small compared to the MGF.

(b) The KL is large compared to the MGF.

Figure 8: Behavior of the gap between Corollary C.3 and the relaxed version of Corollary C.1, when the KL is (a) small or (b) large compared to the moment generating function (MGF) term. A positive value means Corollary C.3 is looser.

## D. Proof and Corollaries of Theorem 2.4

Using Theorem B.1, we can obtain a new sample compression bound for real-valued messages. To do so, we restrict the result to use only Dirac measures as posterior distributions for the compression sets.

**Theorem 2.4** (PAC-Bayes Sample compression - real-valued losses with continuous messages)**.** *For any distribution $\mathcal{D}$ over $\mathcal{X} \times \mathcal{Y}$, for any set $J \subseteq \mathcal{P}(\mathbf{m})$, for any distribution $P_J$ over $J$, for any prior distribution $P_\Omega$ over $\Omega$, for any reconstruction function $\mathcal{R}$, for any loss $\ell : \mathcal{Y} \times \mathcal{Y} \to [0,1]$, for any convex function $\Delta : [0,1] \times [0,1] \to \mathbb{R}$ and for any $\delta \in (0,1]$, with probability at least $1 - \delta$ over the draw of $S \sim \mathcal{D}^m$, we have*

$$\forall \mathbf{j} \in J, Q_\Omega \text{ over } \Omega :$$

$$\Delta\left(\mathop{\mathbb{E}}_{\omega \sim Q_\Omega} \widehat{\mathcal{L}}_{S_{\bar{\mathbf{j}}}}\left(\mathcal{R}(S_\mathbf{j}, \omega)\right), \mathop{\mathbb{E}}_{\omega \sim Q_\Omega} \mathcal{L}_\mathcal{D}\left(\mathcal{R}(S_\mathbf{j}, \omega)\right)\right) \leq \frac{1}{m - \max_{\mathbf{j} \in J} |\mathbf{j}|} \left[\log \frac{1}{P_J(\mathbf{j})} + \mathrm{KL}(Q_\Omega || P_\Omega) + \ln\left(\frac{\mathcal{A}_\Delta(m)}{\delta}\right)\right]$$

*with*

$$\mathcal{A}_\Delta(m) = \mathop{\mathbb{E}}_{(\mathbf{j}, \omega) \sim P} \mathop{\mathbb{E}}_{T_\mathbf{j} \sim \mathcal{D}^{|\mathbf{j}|}} \mathop{\mathbb{E}}_{T_{\bar{\mathbf{j}}} \sim \mathcal{D}^{m - |\mathbf{j}|}} e^{(m - |\mathbf{j}|)\Delta\left(\widehat{\mathcal{L}}_{T_{\bar{\mathbf{j}}}}(\mathcal{R}(T_\mathbf{j}, \omega)), \mathcal{L}_\mathcal{D}(\mathcal{R}(T_\mathbf{j}, \omega))\right)}.$$

*Proof.* We restrict the posterior distributions to be Dirac measures, e.g. for any vector $\mathbf{j}^*$, the posterior distribution is $Q(\mathbf{j}^*) = 1$ and $Q(\mathbf{j}) = 0 \, \forall \mathbf{j} \neq \mathbf{j}^*$. We also separate the KL in two terms.

$$\mathrm{KL}(Q||P) = \mathop{\mathbb{E}}_{h \sim Q} \log \frac{Q(h)}{P(h)}$$

$$= \mathop{\mathbb{E}}_{(\mathbf{j}, \omega) \sim Q} \log \frac{Q(\mathbf{j}, \omega)}{P(\mathbf{j}, \omega)}$$

$$= \mathop{\mathbb{E}}_{\mathbf{j} \sim Q_J} \mathop{\mathbb{E}}_{\omega \sim Q_\Omega} \log \frac{Q(\mathbf{j}, \omega)}{P(\mathbf{j}, \omega)}$$

$$= \mathop{\mathbb{E}}_{\mathbf{j} \sim Q_J} \mathop{\mathbb{E}}_{\omega \sim Q_\Omega} \log \frac{Q_J(\mathbf{j}) Q_\Omega(\omega)}{P_J(\mathbf{j}) P_\Omega(\omega)}$$

$$= \operatorname*{\mathbb{E}}_{\mathbf{j} \sim Q_J} \operatorname*{\mathbb{E}}_{\omega \sim Q_\Omega} \left[ \log \frac{Q_J(\mathbf{j})}{P_J(\mathbf{j})} + \log \frac{Q_\Omega(\omega)}{P_\Omega(\omega)} \right]$$

$$= \operatorname*{\mathbb{E}}_{\mathbf{j} \sim Q_J} \operatorname*{\mathbb{E}}_{\omega \sim Q_\Omega} \log \frac{Q_J(\mathbf{j})}{P_J(\mathbf{j})} + \operatorname*{\mathbb{E}}_{\mathbf{j} \sim Q_J} \operatorname*{\mathbb{E}}_{\omega \sim Q_\Omega} \log \frac{Q_\Omega(\omega)}{P_\Omega(\omega)}$$

$$= \operatorname*{\mathbb{E}}_{\omega \sim Q_\Omega} \left[ Q_J(\mathbf{j}^*) \log \frac{Q_J(\mathbf{j}^*)}{P_J(\mathbf{j}^*)} + \sum_{\mathbf{j} \neq \mathbf{j}^*} Q_J(\mathbf{j}) \log \frac{Q_J(\mathbf{j})}{P_J(\mathbf{j})} \right] + \operatorname*{\mathbb{E}}_{\mathbf{j} \sim Q_J} \operatorname*{\mathbb{E}}_{\omega \sim Q_\Omega} \log \frac{Q_\Omega(\omega)}{P_\Omega(\omega)}$$

$$= \operatorname*{\mathbb{E}}_{\omega \sim Q_\Omega} \left[ 1 \cdot \log \frac{1}{P_J(\mathbf{j}^*)} + \sum_{\mathbf{j} \neq \mathbf{j}^*} 0 \cdot \log \frac{0}{P_J(\mathbf{j})} \right] + \operatorname*{\mathbb{E}}_{\mathbf{j} \sim Q_J} \operatorname*{\mathbb{E}}_{\omega \sim Q_\Omega} \log \frac{Q_\Omega(\omega)}{P_\Omega(\omega)}$$

$$= \log \frac{1}{P_J(\mathbf{j}^*)} + \operatorname*{\mathbb{E}}_{\omega \sim Q_\Omega} \log \frac{Q_\Omega(\omega)}{P_\Omega(\omega)}$$

$$= \log \frac{1}{P_J(\mathbf{j}^*)} + \mathrm{KL}(Q_\Omega || P_\Omega). \qquad \square$$

If we also restrict the posterior distribution on $\Omega$ to be Dirac measures, we obtain a slightly worse version of Theorem 2.3. Indeed, it restricts the comparator function to be convex and it is penalized by the maximum in the denominator and the expectation in $\mathcal{A}_\Delta(m)$, which weren't present in Theorem 2.3.

The following corollaries are easily derived from Theorem 2.4 by choosing a comparator function $\Delta$ and bounding $\mathcal{A}_\Delta(m)$.

**Corollary D.1.** *In the setting of Theorem 2.4, with $\Delta_C(q, p) = -\ln\big(1 - (1 - e^{-C})p\big) - Cq$ (where $C > 0$), with probability at least $1 - \delta$ over the draw of $S \sim \mathcal{D}^m$, we have*

$$\forall \mathbf{j} \in J, Q_\Omega \text{ over } \Omega :$$

$$\operatorname*{\mathbb{E}}_{\omega \sim Q_\Omega} \mathcal{L}_\mathcal{D}(\mathcal{R}(S_\mathbf{j}, \omega)) \leq \frac{1}{1 - e^{-C}} \left[ 1 - \exp\left( -C \operatorname*{\mathbb{E}}_{\omega \sim Q_\Omega} \widehat{\mathcal{L}}_{S_{\bar{\mathbf{j}}}}(\mathcal{R}(S_\mathbf{j}, \omega)) - \frac{\mathrm{KL}(Q_\Omega || P_\Omega) - \ln(P_J(\mathbf{j}) \cdot \delta)}{m - \max_{\mathbf{j} \in J} |\mathbf{j}|} \right) \right].$$

**Corollary D.2.** *In the setting of Theorem 2.4, with $\Delta(q, p) = \mathrm{kl}(q, p) = q \cdot \ln \frac{q}{p} + (1 - q) \cdot \ln \frac{1-q}{1-p}$, with probability at least $1 - \delta$ over the draw of $S \sim \mathcal{D}^m$, we have*

$$\forall \mathbf{j} \in J, Q_\Omega \text{ over } \Omega :$$

$$\mathrm{kl}\left( \operatorname*{\mathbb{E}}_{\omega \sim Q_\Omega} \widehat{\mathcal{L}}_{S_{\bar{\mathbf{j}}}}(\mathcal{R}(S_\mathbf{j}, \omega)), \operatorname*{\mathbb{E}}_{\omega \sim Q_\Omega} \mathcal{L}_\mathcal{D}(\mathcal{R}(S_\mathbf{j}, \omega)) \right) \leq \frac{1}{m - \max_{\mathbf{j} \in J} |\mathbf{j}|} \left[ \mathrm{KL}(Q_\Omega || P_\Omega) + \ln\left( \frac{\mathbb{E}_{\mathbf{j} \sim P_J} 2\sqrt{m - |\mathbf{j}|}}{P_J(\mathbf{j}) \cdot \delta} \right) \right].$$

**Corollary D.3.** *In the setting of Theorem 2.4, for any $\lambda > 0$, with $\Delta(q, p) = \lambda(p - q)$, with a $\varsigma^2$-sub-Gaussian loss function $\ell : \mathcal{Y} \times \mathcal{Y} \to \mathbb{R}$, with probability at least $1 - \delta$ over the draw of $S \sim \mathcal{D}^m$, for all $\mathbf{j} \in J$ and posterior probability distribution $Q_\Omega$, we have*

$$\operatorname*{\mathbb{E}}_{\omega \sim Q_\Omega} \mathcal{L}_\mathcal{D}(\mathcal{R}(S_\mathbf{j}, \omega)) \leq \operatorname*{\mathbb{E}}_{\omega \sim Q_\Omega} \widehat{\mathcal{L}}_{S_{\bar{\mathbf{j}}}}(\mathcal{R}(S_\mathbf{j}, \omega))$$

$$+ \frac{1}{\lambda(m - \max_{\mathbf{j} \in J} |\mathbf{j}|)} \left[ \mathrm{KL}(Q_\Omega || P_\Omega) + \ln \frac{1}{P_J(\mathbf{j}) \cdot \delta} + \ln \operatorname*{\mathbb{E}}_{\mathbf{j} \sim P_J} \exp\left( \frac{(n - |\mathbf{j}|)\lambda^2 \varsigma^2}{2} \right) \right].$$

# E. Proof of Theorem 2.5

**Theorem 2.5** (Disintegrated PAC-Bayes Sample compression bound). *For any distribution $\mathcal{D}$ over $\mathcal{X} \times \mathcal{Y}$, for any set $J \subseteq \mathcal{P}(\mathbf{m})$, for any prior distribution $P_J$ over $J$, for any prior distribution $P_\Omega$ over $\Omega$, for any reconstruction function $\mathcal{R}$, for any compression function $\zeta$, for any loss $\ell : \mathcal{Y} \times \mathcal{Y} \to [0, 1]$, for any $\alpha > 1$, for any convex function $\Delta : [0, 1] \times [0, 1] \to \mathbb{R}$, for any $\delta \in (0, 1]$, with probability at least $1 - \delta$ over the draw of $S \sim \mathcal{D}^m$ (which leads to $(S_{\mathbf{j}^*}, Q_\Omega^S) = \zeta(S, P_\Omega)$ ), and $\omega^* \sim Q_\Omega^S$, we have*

$$\Delta\left( \widehat{\mathcal{L}}_{S_{\bar{\mathbf{j}}^*}}\left(\mathcal{R}(S_\mathbf{j}^*, \omega)\right), \mathcal{L}_\mathcal{D}\left(\mathcal{R}(S_\mathbf{j}^*, \omega^*)\right) \right) \leq \frac{1}{m - \max_{\mathbf{j} \in J} |\mathbf{j}|} \left[ \frac{2\alpha - 1}{\alpha - 1} \ln \frac{2}{\delta} + \ln \frac{1}{P_J(\mathbf{j}^*)} + D_\alpha(Q_\Omega^S || P_\Omega) + \ln \mathcal{A}_\Delta(m) \right].$$

To prove this result, we need the following results from Viallard et al. (2024).

**Theorem E.1** (Viallard et al. 2024). *For any distribution $\mathcal{D}$ on $\mathcal{Z}$, for any hypothesis set $\mathcal{H}$, for any prior distribution $\mathcal{P} \in \mathcal{M}^*(\mathcal{H})$, for any measurable function $\phi : \mathcal{H} \times \mathcal{Z} \to \mathbb{R}_{>0}$, for any $\alpha > 1$, for any $\delta \in (0,1]$, for any algorithm $B : \mathcal{Z}^m \times \mathcal{M}^*(\mathcal{H}) \to \mathcal{M}(\mathcal{H})$, we have*

$$\mathbb{P}_{S \sim \mathcal{D}^m, h \sim \mathcal{Q}_S} \left( \frac{\alpha}{\alpha - 1} \ln \phi(h, S) \right.$$
$$\left. \leq \frac{2\alpha - 1}{\alpha - 1} \ln \frac{2}{\delta} + D_\alpha(\mathcal{Q}_S \,\|\, \mathcal{P}) + \ln \left[ \mathbb{E}_{S' \sim \mathcal{D}^m} \mathbb{E}_{h' \sim \mathcal{P}} \phi(h', S')^{\frac{\alpha}{\alpha - 1}} \right] \right) \geq 1 - \delta,$$

*where $\mathcal{Q}_S = \zeta(S, \mathcal{P})$ is output by the deterministic algorithm $\zeta$.*

*Proof of Theorem 2.5.* Let $\mathcal{Z} = \mathcal{X} \times \mathcal{Y}$. We consider a set of sample-compressed predictors that can be defined using a reconstruction function $\mathcal{R}$, a compression set $\mathbf{j}$ and a message $\omega$. Following from the PAC-Bayes Sample compression work of Laviolette & Marchand (2005), we define the distribution on $J \times \Omega$ instead of the hypothesis class; it is equivalent to sample $h$ from a distribution on the hypothesis class and to sample $(\mathbf{j}^\star, \omega^\star)$ from a distribution on $J \times \Omega$ and set $h = \mathcal{R}(S_{\mathbf{j}^\star}, \omega^\star)$.

Let apply Theorem E.1 with

$$\phi(h, S) = \exp \left( \frac{\alpha - 1}{\alpha} (m - \max_{\mathbf{j} \in J} |\mathbf{j}|) \Delta \left( \widehat{\mathcal{L}}_{S_{\bar{\mathbf{j}}^\star}}(\mathcal{R}(S_{\mathbf{j}^\star}, \omega^\star)), \mathcal{L}_\mathcal{D}(\mathcal{R}(S_{\mathbf{j}^\star}, \omega^\star)) \right) \right).$$

Then, we have

$$\frac{\alpha}{\alpha - 1} \ln \phi(h, S) = \frac{\alpha}{\alpha - 1} \ln \exp \left( \frac{\alpha - 1}{\alpha} (m - \max_{\mathbf{j} \in J} |\mathbf{j}|) \Delta \left( \widehat{\mathcal{L}}_{S_{\bar{\mathbf{j}}^\star}}(\mathcal{R}(S_{\mathbf{j}^\star}, \omega^\star)), \mathcal{L}_\mathcal{D}(\mathcal{R}(S_{\mathbf{j}^\star}, \omega^\star)) \right) \right)$$
$$= \frac{\alpha}{\alpha - 1} \frac{\alpha - 1}{\alpha} (m - \max_{\mathbf{j} \in J} |\mathbf{j}|) \Delta \left( \widehat{\mathcal{L}}_{S_{\bar{\mathbf{j}}^\star}}(\mathcal{R}(S_{\mathbf{j}^\star}, \omega^\star)), \mathcal{L}_\mathcal{D}(\mathcal{R}(S_{\mathbf{j}^\star}, \omega^\star)) \right)$$
$$= (m - \max_{\mathbf{j} \in J} |\mathbf{j}|) \Delta \left( \widehat{\mathcal{L}}_{S_{\bar{\mathbf{j}}^\star}}(\mathcal{R}(S_{\mathbf{j}^\star}, \omega^\star)), \mathcal{L}_\mathcal{D}(\mathcal{R}(S_{\mathbf{j}^\star}, \omega^\star)) \right).$$

Moreover,

$$\ln \left[ \mathbb{E}_{S' \sim \mathcal{D}^m} \mathbb{E}_{h' \sim \mathcal{P}} \phi(h', S')^{\frac{\alpha}{\alpha - 1}} \right]$$
$$= \ln \left[ \mathbb{E}_{h' \sim \mathcal{P}} \mathbb{E}_{S' \sim \mathcal{D}^m} \phi(h', S')^{\frac{\alpha}{\alpha - 1}} \right]$$
$$= \ln \left[ \mathbb{E}_{\mathbf{j} \sim P_J} \mathbb{E}_{\omega \sim P_\Omega} \mathbb{E}_{S'_\mathbf{j} \sim \mathcal{D}^{|\mathbf{j}|}} \mathbb{E}_{S'_{\bar{\mathbf{j}}} \sim \mathcal{D}^{m - |\mathbf{j}|}} \exp \left( \frac{\alpha - 1}{\alpha} (m - \max_{\mathbf{j} \in J} |\mathbf{j}|) \Delta \left( \widehat{\mathcal{L}}_{S'_{\bar{\mathbf{j}}}}(\mathcal{R}(S'_\mathbf{j}, \omega)), \mathcal{L}_\mathcal{D}(\mathcal{R}(S'_\mathbf{j}, \omega)) \right) \right)^{\frac{\alpha}{\alpha - 1}} \right]$$
$$= \ln \left[ \mathbb{E}_{\mathbf{j} \sim P_J} \mathbb{E}_{\omega \sim P_\Omega} \mathbb{E}_{S'_\mathbf{j} \sim \mathcal{D}^{|\mathbf{j}|}} \mathbb{E}_{S'_{\bar{\mathbf{j}}} \sim \mathcal{D}^{m - |\mathbf{j}|}} \exp \left( \frac{\alpha - 1}{\alpha} \frac{\alpha}{\alpha - 1} (m - \max_{\mathbf{j} \in J} |\mathbf{j}|) \Delta \left( \widehat{\mathcal{L}}_{S'_{\bar{\mathbf{j}}}}(\mathcal{R}(S'_\mathbf{j}, \omega)), \mathcal{L}_\mathcal{D}(\mathcal{R}(S'_\mathbf{j}, \omega)) \right) \right) \right]$$
$$= \ln \left[ \mathbb{E}_{\mathbf{j} \sim P_J} \mathbb{E}_{\omega \sim P_\Omega} \mathbb{E}_{S'_\mathbf{j} \sim \mathcal{D}^{|\mathbf{j}|}} \mathbb{E}_{S'_{\bar{\mathbf{j}}} \sim \mathcal{D}^{m - |\mathbf{j}|}} \exp \left( (m - \max_{\mathbf{j} \in J} |\mathbf{j}|) \Delta \left( \widehat{\mathcal{L}}_{S'_{\bar{\mathbf{j}}}}(\mathcal{R}(S'_\mathbf{j}, \omega)), \mathcal{L}_\mathcal{D}(\mathcal{R}(S'_\mathbf{j}, \omega)) \right) \right) \right]$$
$$\leq \ln \left[ \mathbb{E}_{\mathbf{j} \sim P_J} \mathbb{E}_{\omega \sim P_\Omega} \mathbb{E}_{S'_\mathbf{j} \sim \mathcal{D}^{|\mathbf{j}|}} \mathbb{E}_{S'_{\bar{\mathbf{j}}} \sim \mathcal{D}^{m - |\mathbf{j}|}} \exp \left( (m - |\mathbf{j}|) \Delta \left( \widehat{\mathcal{L}}_{S'_{\bar{\mathbf{j}}}}(\mathcal{R}(S'_\mathbf{j}, \omega)), \mathcal{L}_\mathcal{D}(\mathcal{R}(S'_\mathbf{j}, \omega)) \right) \right) \right].$$

With these two derivations, we have a disintegrated PAC-Bayes Sample Compression bound, with a posterior distribution $\mathcal{Q}_S = \zeta(S, P)$. We need to reframe this to accommodate the fact that we want a single compression set and a message sampled from an uncountable set of messages.

To do so, we start by choosing $P = P_J \times P_\Omega$ and $Q_S = Q_J^S \times Q_\Omega^S$. Moreover, we choose $\zeta$ such that $Q_J^S$ is always a Dirac distribution such that the only compression set with a non-zero probability is $\mathbf{j}^\star$. The Rényi Divergence then becomes:

$$D_\alpha(Q\|P) = \frac{1}{\alpha - 1} \ln\left[\mathbb{E}_{(\mathbf{j},\omega)\sim P}\left[\frac{Q(\mathbf{j},\omega)}{P(\mathbf{j},\omega)}\right]^\alpha\right]$$

$$= \frac{1}{\alpha - 1} \ln\left[\mathbb{E}_{(\mathbf{j},\omega)\sim P}\left[\frac{Q_J(\mathbf{j})Q_\Omega(\omega)}{P_J(\mathbf{j})P_\Omega(\omega)}\right]^\alpha\right]$$

$$= \frac{1}{\alpha - 1} \ln\left[\mathbb{E}_{(\mathbf{j},\omega)\sim P}\left[\frac{Q_J(\mathbf{j})}{P_J(\mathbf{j})}\right]^\alpha\left[\frac{Q_\Omega(\omega)}{P_\Omega(\omega)}\right]^\alpha\right]$$

$$= \frac{1}{\alpha - 1} \ln\left[\mathbb{E}_{\omega\sim P_\Omega}\mathbb{E}_{\mathbf{j}\sim P_J}\left[\frac{Q_J(\mathbf{j})}{P_J(\mathbf{j})}\right]^\alpha\left[\frac{Q_\Omega(\omega)}{P_\Omega(\omega)}\right]^\alpha\right]$$

$$= \frac{1}{\alpha - 1} \ln\left[\mathbb{E}_{\omega\sim P_\Omega}\left[\frac{Q_\Omega(\omega)}{P_\Omega(\omega)}\right]^\alpha\mathbb{E}_{\mathbf{j}\sim P_J}\left[\frac{Q_J(\mathbf{j})}{P_J(\mathbf{j})}\right]^\alpha\right]$$

$$= \frac{1}{\alpha - 1} \ln\left[\mathbb{E}_{\omega\sim P_\Omega}\left[\frac{Q_\Omega(\omega)}{P_\Omega(\omega)}\right]^\alpha\left(P_J(\mathbf{j}^\star)\left[\frac{Q_J(\mathbf{j}^\star)}{P_J(\mathbf{j}^\star)}\right]^\alpha + \sum_{\mathbf{j}\neq\mathbf{j}^\star}P_J(\mathbf{j})\left[\frac{Q_J(\mathbf{j})}{P_J(\mathbf{j})}\right]^\alpha\right)\right]$$

$$= \frac{1}{\alpha - 1} \ln\left[\mathbb{E}_{\omega\sim P_\Omega}\left[\frac{Q_\Omega(\omega)}{P_\Omega(\omega)}\right]^\alpha\left(P_J(\mathbf{j}^\star)\left[\frac{1}{P_J(\mathbf{j}^\star)}\right]^\alpha + \sum_{\mathbf{j}\neq\mathbf{j}^\star}P_J(\mathbf{j})\left[\frac{0}{P_J(\mathbf{j})}\right]^\alpha\right)\right]$$

$$= \frac{1}{\alpha - 1} \ln\left[\mathbb{E}_{\omega\sim P_\Omega}\left[\frac{Q_\Omega(\omega)}{P_\Omega(\omega)}\right]^\alpha P_J(\mathbf{j}^\star)\left[\frac{1}{P_J(\mathbf{j}^\star)}\right]^\alpha\right]$$

$$= \frac{1}{\alpha - 1} \ln\left[\mathbb{E}_{\omega\sim P_\Omega}\left[\frac{Q_\Omega(\omega)}{P_\Omega(\omega)}\right]^\alpha\left[\frac{1}{P_J(\mathbf{j}^\star)}\right]^{\alpha-1}\right]$$

$$= \frac{1}{\alpha - 1} \ln\left[\mathbb{E}_{\omega\sim P_\Omega}\left[\frac{Q_\Omega(\omega)}{P_\Omega(\omega)}\right]^\alpha\right] + \frac{1}{\alpha - 1}\ln\left[\left[\frac{1}{P_J(\mathbf{j}^\star)}\right]^{\alpha-1}\right]$$

$$= \frac{1}{\alpha - 1} \ln\left[\mathbb{E}_{\omega\sim P_\Omega}\left[\frac{Q_\Omega(\omega)}{P_\Omega(\omega)}\right]^\alpha\right] + \ln\left[\frac{1}{P_J(\mathbf{j}^\star)}\right]$$

$$= D_\alpha(Q_\Omega\|P_\Omega) + \ln\left[\frac{1}{P_J(\mathbf{j}^\star)}\right]. \qquad\qquad \square$$

## F. Corollaries of Theorem 2.5

**Corollary F.1.** *In the setting of Theorem 2.5, with probability at least $1 - \delta$ over the draw of $S \sim \mathcal{D}^m$, $\mathbf{j} \sim Q_J, \omega \sim Q_\Omega$, we have*

$$\text{kl}\left(\widehat{\mathcal{L}}_{S_{\bar{\mathbf{j}}}}(\mathcal{R}(S_\mathbf{j},\omega)), \mathcal{L}_\mathcal{D}(\mathcal{R}(S_\mathbf{j},\omega))\right) \leq \frac{1}{m - \max_{\mathbf{j}\in J}|\mathbf{j}|}\left[\frac{2\alpha - 1}{\alpha - 1}\ln\frac{2}{\delta} + \ln\frac{1}{P_J(\mathbf{j})} + D_\alpha(Q_\Omega\|P_\Omega) + \ln\mathbb{E}_{\mathbf{j}\sim P_J}2\sqrt{m-|\mathbf{j}|}\right].$$

**Corollary F.2.** *In the setting of Theorem 2.5, with probability at least $1 - \delta$ over the draw of $S \sim \mathcal{D}^m$, $\mathbf{j} \sim Q_J, \omega \sim Q_\Omega$, we have*

$$\mathcal{L}_\mathcal{D}(\mathcal{R}(S_\mathbf{j},\omega)) \leq \frac{1}{1 - e^{-C}}\left[1 - \exp\left(-C\,\widehat{\mathcal{L}}_{S_{\bar{\mathbf{j}}}}(\mathcal{R}(S_\mathbf{j},\omega)) - \frac{1}{m - \max_{\mathbf{j}\in J}|\mathbf{j}|}\left[\frac{2\alpha - 1}{\alpha - 1}\ln\frac{2}{\delta} + \ln\frac{1}{P_J(\mathbf{j})} + D_\alpha(Q_\Omega\|P_\Omega)\right]\right)\right].$$

**Corollary F.3.** *In the setting of Theorem 2.5, for any $\lambda > 0$, with probability at least $1 - \delta$ over the draw of $S \sim \mathcal{D}^m$, $\mathbf{j} \sim Q_J, \omega \sim Q_\Omega$, we have*

$$\mathcal{L}_\mathcal{D}(\mathcal{R}(S_\mathbf{j},\omega)) \leq \widehat{\mathcal{L}}_{S_{\bar{\mathbf{j}}}}(\mathcal{R}(S_\mathbf{j},\omega))$$
$$+ \frac{1}{\lambda(m - \max_{\mathbf{j}\in J}|\mathbf{j}|)}\left[\frac{2\alpha - 1}{\alpha - 1}\ln\frac{2}{\delta} + \ln\frac{1}{P_J(\mathbf{j})} + D_\alpha(Q_\Omega\|P_\Omega) + \ln\mathbb{E}_{\mathbf{j}\sim P_J}\exp\left(\frac{(n-|\mathbf{j}|)\lambda^2\varsigma^2}{2}\right)\right].$$

## G. Algorithm

The following pseudocode depicts our Sample Compression Hypernetworks approach.

---

**Algorithm 1** Training of Sample Compression Hypernetworks (with messages) architecture.

---

**Inputs** : $\mathbf{S} = \{S_i\}_{i=1}^n$, a meta-dataset
$\quad\quad\quad \alpha \in \mathbb{N}$, support set size $(1 \leq \alpha < \min_i[m_i])$
$\quad\quad\quad c, b \in \mathbb{N}$, the compression set and message size
$\quad\quad\quad$ BackProp, a function doing a gradient descent step
$\psi, \phi_1, \phi_2 \leftarrow$ Initialize parameters
**while** Stopping criteria is not met **do**:
$\quad$ **for** $i = 1, \ldots, n$ **do**:
$\quad\quad \hat{S}_i \leftarrow$ Sample $\alpha$ datapoints from $S_i$
$\quad\quad \mathbf{j} \leftarrow \mathcal{C}_{\phi_1}(\hat{S}_i)$ such that $|\mathbf{j}| = c$
$\quad\quad \boldsymbol{\omega} \leftarrow \mathcal{M}_{\phi_2}(\hat{S}_i)$ such that $|\boldsymbol{\omega}| = b$
$\quad\quad \gamma \leftarrow \mathcal{R}_\psi(\hat{S}_{i,\mathbf{j}}, \boldsymbol{\omega})$
$\quad\quad \text{loss} \leftarrow \frac{1}{m_i - \alpha} \sum_{(\mathbf{x},y) \in S_i \setminus \hat{S}_i} l(h_\gamma(\mathbf{x}), y)$
$\quad\quad \psi, \phi_1, \phi_2 \leftarrow$ BackProp(loss)
$\quad$ **end for**
**end while**
**return** $\mathcal{R}_\psi, \mathcal{C}_{\phi_1}, \mathcal{M}_{\phi_2}$

---

## H. Numerical Experiment and Implementation Details

The code for all experiments is available at `https://github.com/GRAAL-Research/DeepRM`.

**Synthetic experiments (moons dataset).** We fixed the MLP architecture in the sample compressor, the message compressor, the reconstructor and the DeepSet modules to a single-hidden layer MLP of size 100 while the predictor also is a single-hidden layer MLP of size 5.

We added skip connections and batch norm in both the modules of the meta-learner and the predictor to accelerate the training time. The experiments were conducted using an NVIDIA GeForce RTX 2080 Ti graphic card.

We used the Adam optimizer (Kingma & Ba, 2015) and trained for at most 200 epochs, stopping when the validation accuracy did not diminish for 20 epochs. We initialized the weights of each module using the Kaiming uniform technique (He et al., 2015).

**Pixels swap and binary MNIST and CIFAR100 experiments.** Let $\text{MLP}_1$ be the architecture of the feedforward network in the sample compressor, the message compressor, the encoder, and the reconstructor; $\text{MLP}_2$ be the architecture of the feedforward network in the DeepSet module; $\text{MLP}_3$ the architecture of the downstream predictor. We used the following components and values in our hyperparameter grid:

- Learning rate: 1e-3, 1e-4;

- $\text{MLP}_1$: [200, 200], [500, 500];

- $\text{MLP}_2$: [100], [200];

- $\text{MLP}_3$: [100], [200, 200];

- $c$: 0, 1, 2, 4, 6, 8;

- $|\boldsymbol{\mu}|, b$: 0, 1, 2, 4, 8, 16, 32, 64, 128.

We added skip connections and batch norm in both the modules of the meta-learner and the predictor to accelerate the training time. The experiments were conducted using an NVIDIA GeForce RTX 2080 Ti graphic card.

We used the Adam optimizer (Kingma & Ba, 2015) and trained for at most 200 epochs, stopping when the validation accuracy did not diminish for 20 epochs. We initialized the weights of each module using the Kaiming uniform technique (He et al., 2015).

Table 5: Comparison of different meta-learning methods and generalization bounds on the MNIST and CIFAR100) binary task. The 95% confidence interval is reported for generalization bound and test error, computed over the test tasks.

| Algorithm | MNIST | | CIFAR100 | |
|---|---|---|---|---|
| | Bound ($\downarrow$) | Test error ($\downarrow$) | Bound ($\downarrow$) | Test error ($\downarrow$) |
| PBH | $0.597^* \pm 0.107$ (Thm. 2.1) | $0.150 \pm 0.114$ | $0.974^* \pm 0.022$ (Thm. 2.1) | $0.295 \pm 0.103$ |
| | $0.770 \pm 0.076$ (Thm. 2.5) | " | $0.999 \pm 0.001$ (Thm. 2.5) | " |
| SCH$_-$ | $0.352 \pm 0.187$ (Thm. 2.2) | $0.278 \pm 0.076$ | $0.600 \pm 0.143$ (Thm. 2.3) | $0.374 \pm 0.118$ |
| | $0.369 \pm 0.187$ (Thm. 2.3) | " | $0.629 \pm 0.142$ (Thm. 2.5) | " |
| SCH$_+$ | $0.280 \pm 0.148$ (Thm. 2.2) | $0.155 \pm 0.109$ | $0.745 \pm 0.101$ (Thm. 2.2) | $0.305 \pm 0.142$ |
| | $0.295 \pm 0.148$ (Thm. 2.3) | " | $0.758 \pm 0.098$ (Thm. 2.5) | " |
| PB SCH | $0.597^* \pm 0.107$ (Thm. 2.4) | $0.150 \pm 0.114$ | $0.974^* \pm 0.022$ (Thm. 2.4) | $0.295 \pm 0.103$ |
| | $0.770 \pm 0.076$ (Thm. 2.5) | " | $0.999 \pm 0.001$ (Thm. 2.5) | " |

$^*$Bound on average over the decoder output.

**More results on binary MNIST and CIFAR100 experiments.** We present in Figure 9(a) and Table 6 (Figure 9(b) and Table 7) the contribution of each of the terms impacting the bound value for a few algorithms on the MNIST 200 pixels swap (binary CIFAR100) task. In the figures, the cumulative contributions are displayed, while in the tables, the marginal contributions are displayed. The bounds are decomposed as follows:

- The observed meta train "error";

- The "confidence penalty", which corresponds to the term $-\ln\theta$ in Theorem 2.1 and the corresponding term in other bounds;

- The "complexity term", which corresponds to the KL factor in the PAC-Bayes bounds. The latter is further decomposed into the compression set probability and the message probability in our sample compression-based bounds.

When considering the decomposition on the 200 pixels swap experiment, we see that our approaches, despite having a larger error term, relies on a small message probability and an empty compression set to yield competitive bounds. In contrast, for Pentina & Lampert (2014), the complexity term profoundly impacts the bound, making it non-competitive. As for the decomposition on the CIFAR100 experiments, it is interesting to see that the bound from Zakerinia et al. (2024) and the one from PB SCH have a similar decomposition, whereas SCH$_+$, despite being penalized by the message probability, relies on a better treatment of its error and confidence term to obtain best bound of the four considered algorithms. This is empirical evidence of the tightness of our bounds compared to those of the runner-ups, all factors (error, confidence $\theta$, ...) being kept equal, thanks to the non-linear comparator function $\Delta$ (see Theorem 2.1).

**Label shuffle experiment.** As suggested by one ICML reviewer, we performed an additional experiment, based on the one in Amit & Meir (2018) (as well as Zakerinia et al. (2024)). The *label shuffle* experiment goes as follows: we start with the MNIST multiclass dataset, and we create each new task by performing random permutations of the label space. There are 30 training tasks of 1000 examples, and we evaluated the methods on 10 tasks of 100 samples. Table 8 presents the obtained results. We witness that, despite an extensive grid search over hyperparameters, the PB SCH approach fails to learn to generalize to the test tasks. We empirically observed severe overfitting on meta-train tasks. We hypothesize that this is due to the DeepSet architecture, which might not be powerful enough to encode the subtleties of the tasks at hand.

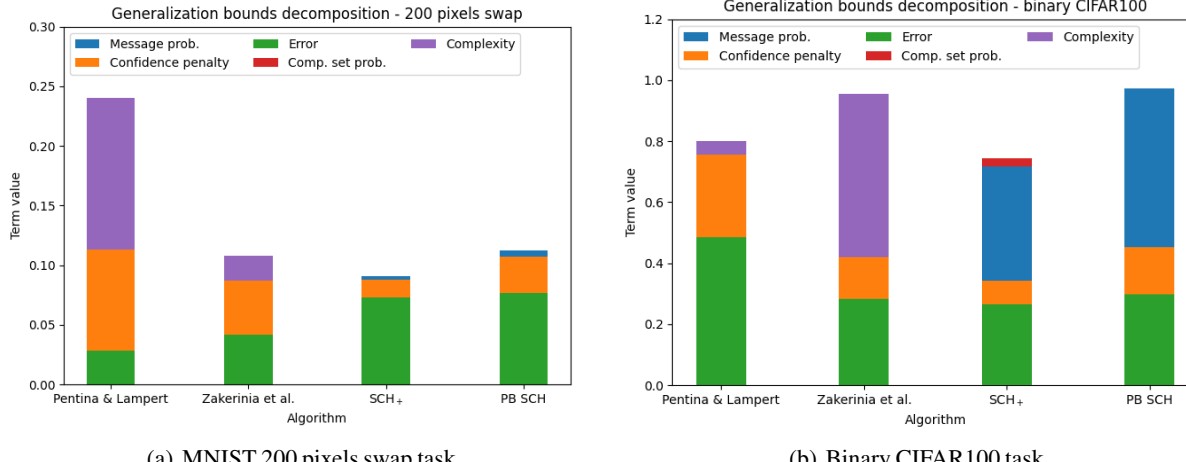

(a) MNIST 200 pixels swap task  (b) Binary CIFAR100 task

Figure 9: Bound values cumulative decomposition for a few algorithms.

Table 6: Bound value decomposition for a few algorithms on the 200 pixels swap task.

| Algorithm | Error | Confidence penalty | Complexity | | Total |
|---|---|---|---|---|---|
| (Pentina & Lampert, 2014) | $0.028 \pm 0.002$ | $0.085 \pm 0.000$ | $0.127 \pm 0.028$ | | $0.240 \pm 0.030$ |
| (Zakerinia et al., 2024) | $0.042 \pm 0.198$ | $0.045 \pm 0.000$ | $0.021 \pm 0.160$ | | $0.108 \pm 0.037$ |
| | Error | Confidence penalty | Message prob. | Comp. set prob. | Total |
| SCH$_+$ | $0.073 \pm 0.017$ | $0.015 \pm 0.002$ | $0.003 \pm 0.001$ | $0.000 \pm 0.000$ | $0.091 \pm 0.022$ |
| PB SCH | $0.077 \pm 0.016$ | $0.030 \pm 0.001$ | $0.005 \pm 0.005$ | $0.000 \pm 0.000$ | $0.112 \pm 0.021$ |

Table 7: Bound value decomposition for a few algorithms on the binary CIFAR100 task.

| Algorithm | Error | Confidence penalty | Complexity | | Total |
|---|---|---|---|---|---|
| (Pentina & Lampert, 2014) | $0.485 \pm 0.068$ | $0.270 \pm 0.000$ | $0.046 \pm 0.069$ | | $0.801 \pm 0.001$ |
| (Zakerinia et al., 2024) | $0.283 \pm 0.120$ | $0.133 \pm 0.000$ | $0.537 \pm 0.195$ | | $0.953 \pm 0.315$ |
| | Error | Confidence penalty | Message prob. | Comp. set prob. | Total |
| SCH$_+$ | $0.264 \pm 0.133$ | $0.078 \pm 0.010$ | $0.374 \pm 0.017$ | $0.027 \pm 0.003$ | $0.745 \pm 0.101$ |
| PB SCH | $0.298 \pm 0.098$ | $0.155 \pm 0.008$ | $0.521 \pm 0.045$ | $0.000 \pm 0.000$ | $0.974 \pm 0.022$ |

Table 8: Comparison of different meta-learning methods on the MNIST label shuffle binary task. The 95% confidence interval is reported for generalization bound and test error, computed over 10 test tasks. The best (smallest) result in each column is **bolded**.

| Algorithm | MNIST label shuffle | |
|---|---|---|
| | Bound ($\downarrow$) | Test error ($\downarrow$) |
| (Pentina & Lampert, 2014) | $2.376 \pm 0.001$ | $0.900 \pm 0.589$ |
| (Amit & Meir, 2018) | $0.542 \pm 0.034$ | $\mathbf{0.023} \pm 0.062$ |
| (Guan & Lu, 2022) - kl | $3.199 \pm 0.372$ | $\mathbf{0.023} \pm 0.006$ |
| (Guan & Lu, 2022) - Catoni | $\mathbf{0.536} \pm 0.063$ | $0.030 \pm 0.001$ |
| (Rezazadeh, 2022) | $12.95 \pm 0.001$ | $0.902 \pm 0.660$ |
| (Zakerinia et al., 2024) | $3.779 \pm 0.079$ | $0.165 \pm 0.023$ |
| PB SCH | $0.997^* \pm 0.005$ | $0.792 \pm 0.070$ |

$^*$Bound on average over the decoder output.

