# OpenReview forum: "Generalization Bounds via Meta-Learned Model Representations: PAC-Bayes and Sample Compression Hypernetworks"
_ICML.cc/2025/Conference — ICML 2025 poster_

### Official Review · Reviewer_KTTB · 2025-03-13

**Overall Recommendation:** 3

**Summary:**

--- increased score from 2 to 3 after comment from authors ---

The authors developed a sample compression version of PAC Bayes generalization bounds, which reduce the number of training data points in standard PAC Bayes bounds into a compressed subset with generalization guarantee. They used a hypernetwork to meta-learn the parameter of another network, which they applied the PAC Bayes bound to. The authors explore three different hypernetwork architectures. Specifically, the authors design the sample compression hypernetwork to encode both samples and messages and decode the parameters of a downstream predictor. For hybrid networks, the message encoder is replaced by a PAC Bayes encoder. They evaluate their method over synthetic tasks and image tasks like MNIST variants and CIFAR100. The proposed bounds and training procedure lead to a tighter bound compared to a couple of baselines.

**Claims And Evidence:**

Looks reasonable to me. There are no signs of extravagant claims.

**Essential References Not Discussed:**

Missing https://arxiv.org/abs/2407.18158, https://arxiv.org/abs/2312.17173. PAC Bayes bounds on LLMs. The proposed subspace compression via a linear mapping is similar to the hypernetwork idea here.

**Experimental Designs Or Analyses:**

--- after authors' responses: These are promising avenues for future work indeed. Thank you for answering and I don't expect results on this since it would be beyond the scope of this paper. ---

I’m a bit skeptical about why meta-learning the parameters of a downstream predictor is a good idea. Would this meta-learning be too restrictive for very large models? How does this meta-learning approach compared to other types of model compression like subspace methods https://arxiv.org/abs/1804.08838, quantization https://arxiv.org/abs/2407.18158 etc. Why is the hypernetwork approach potentially better than other approaches listed above. Would be great to motivate this or demonstrate the superiority of this approach empirically.

**Methods And Evaluation Criteria:**

The benchmark dataset (MNIST, CIFAR100) is quite standard, though they are a bit small scale. Nowadays people are applying bounds on models like 100M~7B parameters. See https://arxiv.org/abs/2407.18158, https://arxiv.org/abs/2312.17173.

**Other Comments Or Suggestions:**

The paper can benefit from more analysis figures. For example, I would like to know the contribution of each of the terms in the bound to the bound value. Might be interesting to plot it out and visualize which part takes up most of the bound value when summed together.

**Other Strengths And Weaknesses:**

Strength:
- the combination of PAC Bayes and sample compression is novel. The hypernetwork design here is novel.
- bounds derivation looks reasonable.

--- after authors' responses: Thank you for explaining the differences between these two tables. It makes more sense now. ---

Weakness:
- insufficient motivation for hypernetworks
- insufficient empirical demonstration for hypernetworks, though this is a minor point depending on how much compute the authors have.
- empirical results in Table 1 and 2 are a bit weak. Some of the bound values are higher than the strawman baseline.

**Questions For Authors:**

as shown above.

**Relation To Broader Scientific Literature:**

This paper is an extension of a line of works in PAC Bayes bounds. More specifically, the authors combined the KL term in PAC Bayes and the term introduced by sample compression to bound generalization error.

**Theoretical Claims:**

I’m not an expert on sample compression bounds, but the combinations of PAC Bayes and sample compression bounds look reasonable to me. The KL term and the terms introduced by sample compression are within expectations, though I didn’t verify every details in the derivation.

---

> ### Author Rebuttal · Authors · 2025-04-01
>
> We thank the reviewer for his insightful feedback.
>
> **1.** “People are applying bounds on models like 100M~7B parameters.”
>
> Indeed, interesting works have successfully computed tight generalization bounds for large models. It does not undermine the need for tighter generalization bounds for smaller models, as they are still used for many applications. Moreover, the submitted manuscript focuses on investigating a new way of obtaining bounds in meta-learning, which we conceive as a stepping stone for undertaking larger experiments.
>
> **2.** “Missing literature review on PAC Bayes for LLMs”
>
> Following the reviewer's comments, we will include a literature review of PAC-Bayes for LLMs in Section 2.1 and mention these two works: “Unlocking Tokens as Data Points for Generalization Bounds on Larger Language Models” and “Non-Vacuous Generalization Bounds for Large Langage Models”.
>
> **3.** “The proposed subspace compression is similar to the hypernetwork idea [ in SubLoRA (https://arxiv.org/abs/2312.17173) ].”
>
> We would like to emphasize the following differences between SubLoRA and our approach: the representation in SubLoRA (and in LoRA) is learned via SGD and is composed of tunable parameters. In our case, though the representation is a function of tunable parameters, it mostly depends on the input dataset. The task is not to encode the weights of a model working well on a single task but to ensure the versatility of the representation for any related task. The way the representation is treated is also quite different: in our case, it is fed to a hypernetwork whose output is the weights of the downstream predictor. (See also **6.** for a conceptual difference.)
>
> **4.** “Empirical results in Tables 1 and 2 are a bit weak. Some of the bound values are higher than the strawman baseline.”
>
> We report empirical evaluation in both unfavorable (Table 1) and favorable (Table 2) environments. Table 1 illustrates that a fixed model can perform well across the tasks in terms of both error and bound. This highlights that the commonly used pixel swap environment does not account for all meta-learning scenarios. See “Rebuttal XmYz, 2.2” for a numerical investigation.
>
> The strawman bounds are essentially test bounds: they are valid for a single predictor. Since no predictor is learned on the query set, the empirical error is an unbiased test error estimate (close to 0.5 in Table 2). Hence, the gap between the bounds and the test error is small in these cases. In contrast, the train bounds underlying the other methods are valid uniformly for all learnable predictors. The price to obtain train bounds is a larger complexity term that increases the gap. For the CIFAR100 experiment, all training bounds are greater than the one of the baseline, but we achieved the lowest one with our SCH- model.
>
> **5.** “I’m a bit skeptical about why meta-learning the parameters of a downstream predictor is a good idea. Too restrictive for very large models?”
>
> Since our bound relies on the latent representation instead of the complexity of the downstream predictor and decoder, we expect it to be suited for fine-tuning of the last few layers of large models. Also, one could consider a “prior” over the downstream predictor, which would correspond to the random initialization in the LoRA nomenclature, and predict the weights of LoRA-like matrices to modify this “prior”. Finally, note that our bounds hold for any bounded loss function, which is necessary in some settings for bounding models generating sequences of tokens. These are all promising future works that we will mention in our conclusion.
>
> **6.** “How does this meta-learning approach compare to other types of model compression like [...]?”
>
> Although both our approach and the aforementioned methods are compression-based, ours is not a model-compression approach. Our primary goal is to compress the dataset into a subset of datapoints and a message, which is a less explored strategy. By creating a bottleneck, we compress the information contained in the dataset and use it to learn a model instead of compressing the model itself. Therefore, to our knowledge, we compared ourselves to the works closest to ours in the field of meta-learning: PAC-Baysian approaches.
>
> **7.** “The paper can benefit from more analysis figures [...]."
>
> During the rebuttal period, we crafted additional figures following the Reviewers' suggestions:
> - We present the contribution of each of the terms in the bound to the bound value for a few algorithms on the 200 pixels swap (see https://imgur.com/a/9d4fOvB) and on the CIFAR100 binary task (see https://imgur.com/a/GSyoggu). See “Rebuttal to Reviewer 5RkH, 2.4” for a discussion on these figures.
> - We depict the test error and generalization bound for PB SCH as a function of both the compression set size and the message size. (see https://imgur.com/a/r45Wq56 and “Rebuttal to Reviewer 5RkH, 2.3” discussion).

---

> > ### Comment · Reviewer_KTTB · 2025-04-02
> >
> > Dear authors, thank you for answering my questions and providing additional figures! It would be helpful to include discussions with related work for better context. I'm still somewhat skeptical about the empirical performance similar to Reviewer XmYZ. But in light of the novelty in the proposed method and addressed concerns, I'm raising my score from 2 to 3.

---

> > > ### Author Response · Authors · 2025-04-03
> > >
> > > We are grateful to the reviewer for assessing the novelty of our contribution! We will add the new figures, discussions, and references. Thanks to the Reviewers' comments, these will undoubtedly improve the manuscript.

---

### Official Review · Reviewer_XmYZ · 2025-03-13

**Overall Recommendation:** 2

**Summary:**

This paper proposes  a novel meta-learning framework that uses PAC-Bayes and Sample compression theory to learn the hypernetwork parameters.  The hypernetwork consists of two components: an encoder (or compressor) that maps the training set into the latent representation space, and a decoder (or reconstructor) the maps the latent representation into the parameter of downstream predictor. A new PAC-Bayes sample compression bound as well as it disintegrated variant are provided to guarantee the low generalization error of the posterior over the latent representation space. Experiments are also conducted to validate the effectiveness of proposed method.

**Claims And Evidence:**

Weaknesses:

**(1) My first major concern is that, the theorectical results are not novel enough.** Two reasons are as follow:

(i) Theorems 2.1-2.3 are the existing results. Although authors may place Theorems 2.1-2.3 in the main body for the completeness of the whole work, some of them in my opinion can be deferred to the supplementary material (e.g. Theorem 2.2. can be somewhat regarded as a special result of Theorem 2.3 and did not be used very few times in the main body, hence can be deferred to the appendix).

(ii) The proof technique is not new. The proof for the disintegrated bound in Theorem 2.5 is actually a direct extension of Theorem E.1 by Viallard et al. 2024. The proof technique (i.e. the use of change of measure lemma, the Markov's inequality, as well as the bounding of moment generating function of the comparator function $\Delta$) of Theorems B.1 and  B.2 (i.e. of Theorem 2.4) is basic, and the contribution is to integrate PAC-Bayes bound into the sample compression framework, which is not new for me and seems not technical enough.

----

**(2) The second major concern is that, the experimental results in Tables 1-2 of the proposed method are not convincing .**  Several reasons are as follow:

(i) First, in Table 1, the proposed method could not outperform ALL PAC-Bayes meta learning methods in terms of the test error (although the proposed method can achieve the tightest bounds). The explanation for this phenomenon lies in line 380 right column that, "The benchmark methods perform well because the posterior for each test task, is similar to the prior." As far as I know, the posterior for test tasks in the benchmark methods (like (pentina 2014 icml)) is outputed by running the PAC-Bayes algorithm that takes prior (sampled from the informative hyper-posterior) and the training data as input. Then, how to demonstrate the posterior and the prior is similar? If they are similar, then the KL-divergence between them will be small and the PAC-Bayes bound (of benchmark method) is small, then why the test bounds of the proposed method in Table 1 is smaller than the test bound of benchmark method?

(ii) In Table 1 and CIFAR100 of Table 2, the performance PBH and PB SCH is worse than the propoesd SCH _ and SCH +. And even worse is that, PBH and PB SCH could not outperform many benchmark methods in terms of the tightness of test bound the test error. Then it seems that the proposed PAC-Bayes encoder-decoder hypernetworks in Section 3.1 is not good enough. In particular, given that PB SCH is the hybrid version of PBH and SCH but performs much worse than SCH, I am not convinced that the proposed PAC-Bayes encoder-decoder hypernetworks (and it variant PB SCH) works well.

**Essential References Not Discussed:**

Yes.

**Experimental Designs Or Analyses:**

The proposed meta-learning method uses support-query training strategy that needs to split the training dataset into two non-overlapping parts. But according to my experience, the benchmark methods in Table 1 \&2 (like [PL2014], [AM2018],[GL2022]) uses traditional ERM training strategy, instead of such support-query strategy. Then, in the experimental setting, for example each training task including 2000 images, benckmark methods compute the training loss over 2000 images but the proposed method only compute the training loss over part of these 2000 images?

**Methods And Evaluation Criteria:**

Yes.

**Other Comments Or Suggestions:**

Typos:

(1) line 91: by many (investigators)

(2) line 105: asses->assess

(3) line 122: abbreviation SCM lacks explanation

(4) line 186, right column: $\mathscr{H}(S') ->\mathscr{H}_{\theta}(S') $

**Other Strengths And Weaknesses:**

Strengths:

(1) The new PAC-Bayes sample compression bound in Theorem 2.4 is well-motivated by existing works in Theorems 2.1-2.3.

(2) Experimental results in Table 2 validate the effectivess of the proposed hypernetwork framework.

**Questions For Authors:**

(1) In the experimental setting, for example each training task including 2000 images, benckmark methods compute the training loss over 2000 images but the proposed method (with support-query strategy) only compute the training loss over part of these 2000 images?

(2) What is the main technical difficulty of combining PAC-Bayes bound and sample compression scheme to obtain new bound in Theorem 2.4?

**Relation To Broader Scientific Literature:**

This paper provides a new PAC-Bayes meta-learning framework to analyze the generalization performance of predictor on downstream tasks, which is different from the existing literature that considered the hierarchy of prior and posterior distribution.s

**Theoretical Claims:**

I check the proof as far as I can, and believe it is correct. However, some crucial explanations should be added to make the whole proof clearer and more readable. For example, in line 1069,  why "${\mathbb{E}}\left[\frac{Q_J(\mathbf{i})}{P_J(\mathbf{i})}\right]^\alpha=P_J(\mathbf{j})\left[\frac{1}{P_J(\mathbf{j})}\right]^\alpha$" holds? Such kind of explanations should be added.

---

> ### Author Rebuttal · Authors · 2025-04-01
>
> We thank the reviewer for his feedback, which will help us highlight the precise nature of our contribution. We undertake to add these clarifications to the manuscript.
>
> **1. First concern (the theoretical results are not novel enough)**
>
> **1.1.** We agree that our theoretical results are moderately novel, and we respectfully suggest that our work should not be judged on this basis. Our main contributions are not new PAC-Bayesian and sample compression generalization bounds or advanced proof techniques. Instead, we propose an original way to leverage existing results in a meta-learning framework. We are the first to apply PAC-Bayesian and sample compression to hypernetworks. With this in mind, we undertake to present various kinds of generalization bounds in a unified setting, advancing the idea that each bound can inspire a particular hypernetwork architecture. To our knowledge, this unconventional perspective is fresh, and we argue that it deserves to be shared with the machine learning community.
>
> **1.2.** Hence: “What is the main technical difficulty of [...] to obtain new bound in Theorem 2.4?” There is no technical difficulty here for those being familiar with the usual PAC-Bayes and sample compress proof schemes; our result is indeed based on existing ones. The interest in our results does not lie in their complexity. In the proof of Theorem B.1 and Theorem B.2, we unify and generalize the proof techniques of previous results. Doing so, we present Theorem B.1, which recovers a version of the theorem of Laviolette & Marchand (2005) for real-valued losses and a version of the theorem of Thiemann et al. (2017) for different sizes of compression sets. Theorem B.2 is a tighter version of previous works and a correct version of Theorem 39 of Germain et al. (2015). Indeed, as pointed out in lines 809-839, their proof was only correct for one size of compression set.
>
> Finally, Theorem 2.4 is a direct consequence of Theorem B.1. The idea of using a Dirac distribution to obtain “sample-compression” style bounds is not novel, as it was explored in Marchand & Laviolette (2005). Nevertheless, mixing probabilistic messages and deterministic compression sets is entirely new and follows our peculiar way of looking at generalization bounds to inspire original hypernetworks. This enables us to use real-valued messages instead of binary messages, which is pivotal to optimizing the parameters of the PB SCH architecture bounds with respect to the message in a continuous way.
>
> **2. Second concern (the experimental results are not convincing)**
>
> **2.1.** We rigorously reported the empirical evaluation in both unfavorable (Table 1) and favorable (Table 2) environments. On the binary MNIST task, our methods compare very favorably to others. On the CIFAR100 task, our method achieves the best bound by a reasonable margin.
>
> **2.2.** The reviewer asks “how to demonstrate the posterior and the prior is similar?” To better illustrate our claim, we computed the prior-posterior KL term of both Pentina & Lampert (2014) and Zakerinia et al. for a few tasks (see https://imgur.com/a/tcIJlNZ). This shows that the KL values are lower for a small amount of pixel swap and larger for the binary MNIST task.
>
> **2.3.** As to “why the test bounds in Table 1 are smaller than the test bound of benchmark methods”, it partly relies on the fact that our bounds are based on the $kl(q,p)$ comparator function (see Eq. 1) instead of being linear as in Pentina (2014). Also, a small compression set and message are selected; thus, only a small complexity term enters the computation of the bound.
>
> **2.4.** Concerning the performances of PB SCH, we study in a new figure (see https://imgur.com/a/r45Wq56) the test error and generalization bound for PB SCH as a function of both the compression set size and the message size. See “Rebuttal to Reviewer 5RkH, 2.3” for a discussion of these results.
>
> **3. “Theorems 2.1-2.3 are existing results [... and] can be deferred to the supplementary material.”**
>
> We thank the reviewer for the suggestion. We will move these parts to the appendix and use the freed space to clarify the abovementioned points.
>
> **4. “In the experimental setting, each training task including 2000 images [...] of these 2000 images?”**
>
> This is right: half the images (the support set) are used to generate the predictor, while the other half (the query set) is used to compute the meta train loss. However, a new random support/query split is performed at each epoch for every train dataset.
>
> **5. “Some crucial explanations should be added to make the whole proof clearer and more readable. For example, in line 1069, why [...].”**
>
> On Line 1069, the expectation is on discrete distributions  $P_J$, and $Q_J$, the latter being a Dirac on $\mathbf{j}$ such that $Q(\mathbf{j})=1$, explaining the equality. We will add this clarification and comment on the many other steps of the proof.

---

> > ### Comment · Reviewer_XmYZ · 2025-04-02
> >
> > Thank authors for the detailed responses. Some of my concerns have been addressed.  My responses are as follow:
> >
> > (1) I still value the theoretical contribution of one paper submitted to ICML conference. As the authors acknowledged, only Theorems 2.4 and 2.5 are the novel results in this paper, and the proof technique is not new and not technical. This to some extent will lower the theoretical novelty of this paper.
> >
> > Nevertheless, authors also suggest that the propoed hypernetworks based on PAC-Bayes and sample compression is the key contribution of this paper. I agree with this point, and the proposed method is novel for me.
> >
> > (2) For the second concern, the performance of PB SCH (i.e. the combination of PAC-Bayes and sample compression) is not satisfactory. I think as the combation, PB SCH can outperform both PBH and SCH, but in Table 1 and 2 SCH obtains better performance. The new figure seems to show that a better tradeoff combination of PBH and SCH can lead to a better result, but in practice it is hard to tune the trade-off parameter. For this concern, I still doubt that whether PBH this method is effective, when compared with other benchmark methods and compared with SCH. If PBH does not work well (i.e. at least comparable results with other in Table 1 and 2) in most datasets, the effectiveness of the proposed  hypernetwork is not sound enough.
> >
> > Overall, considering the effectiveness of the proposed algorithms as well as the theoretical contribution, i will maintain the initial score at the current stage.

---

> > > ### Author Response · Authors · 2025-04-03
> > >
> > > We thank the reviewer for the well-argumented response.
> > >
> > > We agree that the trade-off of PBSCH is challenging to tune. This is why we initially performed the model selection relying on a validation set in Tables 1 and 2, which end up selecting the same models for PB SCH and PBH (without any compression set). The new figure (https://imgur.com/a/r45Wq56) indeed shows the tradeoff between the message size and the compression set size. We will include this figure in the revised version of our manuscript, as it helps in grasping the behavior of the investigated hypernetworks, and we will also honestly mention the difficulty of performing model selection based on the bounds as one of the current limitations of our approach.
> > >
> > > While we disagree that PBH does not work well in the MNIST/CIFAR binary environments, we intentionally did not proclaim that one version of PBH, SCH or PBSCH is the most effective. We proposed, investigated and compared three possibilities. Among them, PBSCH is a hybrid of PBH and SCH, and comes with its own challenge due to the increased complexity. These three examples may serve as a starting point for the community to explore generalization bounds in this original setting, e.g., bounds and architectures suitable to LLM-scale models by reconstructing the parameters of LoRA layers.

---

### Official Review · Reviewer_5RkH · 2025-03-14

**Overall Recommendation:** 4

**Summary:**

The paper introduces new generalization bounds combining both PAC-Bayes and sample compression framework, and apply it in a meta-learning scheme. They introduce three different designs inspired by different theorems by using hypernetworks.

**Claims And Evidence:**

Technically, the generalization bounds proved in this paper are not meta-learning generalization bounds. Meta-learning bounds, usually are applied in the Baxter's setting during the meta-learning *training* phase based on the training task, to guarantee the performance of the future test tasks. The bounds provided here however, assume that the training phase is over and the hypernetworks are learned from previous tasks, and provide a bound on the generalization gap for the test tasks. I think this should be made clear in the paper, in discussion to related works. Other than this, the paper is clear.

**Essential References Not Discussed:**

The paper uses hypernetworks, however does not have any citations related to it or any prior works who use hypernetworks. The following papers are some relevant references which probably should be discussed. The first paper is the paper that introduces the hypernetworks, and the other three use hypernetworks for personalized federated learning, in a meta-learning scheme, which I think are relevant. The use of hypernetwork is similar to the current paper, and a discussion and comparison is needed.

- Ha, D., Dai, A. M., and Le, Q. V. HyperNetworks. ICLR, 2017.

- Shamsian, A., Navon, A., Fetaya, E., and Chechik, G. Personalized Federated Learning using Hypernetworks. ICML, 2021.

- Scott, J., Zakerinia, H., and Lampert, C. H. PeFLL: Personalized Federated Learning by Learning to Learn. ICLR, 2024.

- Amosy, O., Eyal, G., and Chechik, G. Late to the party? On-demand unlabeled personalized federated learning. WACV, 2024.

**Experimental Designs Or Analyses:**

- As mentioned above, the prior works provide a meta-learning bound in the training phase, and this work is a transfer bound. How did you do the comparison of the tables? Are they all test time bounds? And yes, can you clarify how did you compute the bounds for the baselines?

- Amit & Meir, 2018 had another experiments based on MNIST in which they shuffle the labels instead of pixels. That experiment can be interesting, since I assume sample compression would need at least 10 samples. It is interesting to see if it is possible to get good result with less than 10 samples by using the message.

- The new bound proved in the paper is about PB SCH, however, in the experiments, it has $c=0$ and it is reduced to PB setting. Is it because additional samples doesn't improve? Can you come up with a setting, which $c>0$ helps?

**Methods And Evaluation Criteria:**

The provided theorem and the methods derived from them are interesting, and the addition of the sample compression for a meta-learning setting is novel. In general the experiments are done based on standard benchmarks as in prior works, and the results are good. However, I have the following questions/concerns. See Below.

**Other Comments Or Suggestions:**

- I think the pixel swap experiments when there are less samples per task for training tasks and/or test tasks is more interesting. I assume Opaque encoder does not work in this setting.

**Other Strengths And Weaknesses:**

_

**Questions For Authors:**

- Why are some bounds in table 2 bigger than 1? especially, there is a 1372 in the table. Is this correct?

**Relation To Broader Scientific Literature:**

Prior works focus on the generalization gap in the meta-learning setting, however the focus of this paper is improving the generalization gap for a given task, after the meta-learning training. Hypernetworks were also used in the prior works for meta-learning, but the addition of sample compression is novel and an  interesting idea for gain improvement.

**Theoretical Claims:**

I skimmed the proofs and they seem correct, however I didn't check them in full detail.

---

> ### Author Rebuttal · Authors · 2025-04-01
>
> We thank the reviewer for his careful reading of the paper.
>
> **1. Claims And Evidence**
>
> The reviewer correctly says that “the generalization bounds proved in this paper are not meta-learning generalization bounds.” Instead, our framework shows a new way of using generalization bounds in a meta-learning framework. We will clarify this further to avoid misconceptions, especially in the Related Works part of the introduction.
>
> **2. Experimental Designs Or Analyses:**
>
> **2.1.** Concerning the comparison with the benchmarks: for our algorithms, we naturally used the support set for the computation of the bound, and the query set for the computation of the test error. Though the benchmarks do not require support/query splits, it is necessary to compute the bound and the test error on independent sets to ensure the reliability of the results. Since the query/support split is done in a random 50/50 fashion for our approaches, we split similarly the datasets for the benchmarks to compute the bound and the test error.
>
> **2.2** Concerning the labels shuffle experiment, we agree it would be an interesting use case for our algorithms and will add this experiment to the revised version of our manuscript.
>
> **2.3** Concerning PB SCH and  $c > 0$: we produce new results (see: https://imgur.com/a/r45Wq56) depicting the test error and generalization bound for PB SCH as a function of both the compression set size and the message size. We see that the message is better-suited for the minimization of the test error, but a trade-off between the compression set size and the message size is required to obtain the best bounds (In Table 1 and 2, we used the validation error to select the compression set size and message size). Interestingly, when we enforce a small message size, the benefit of using $c>0$ becomes apparent.
>
> **2.4** We also present the contribution of each of the terms in the bound to the bound value for a few algorithms on the 200 pixels swap (see here: https://imgur.com/a/9d4fOvB) and on the CIFAR100 binary task (see here: https://imgur.com/a/GSyoggu). In the figures, the cumulative contributions are displayed, while in the tables, the marginal contributions are displayed. The bounds are decomposed as follows:
> - The observed meta train “error”
> - The “confidence penalty”, which corresponds to the term $-\ln(\delta)$ in Theorem 2.1 and the similar terms in other bounds
> - The “complexity term”, which corresponds to the KL factor in the PAC-Bayes bounds. The latter is further decomposed into the compression set probability and the message probability in our sample compression-based bounds.
>
> When considering the decomposition on the 200 pixels swap experiment, we see that our approach, despite having a larger error term, relies on a small message probability and a null compression set probability to yield competitive bounds. In contrast, for Pentina & Lampert, the complexity term profoundly impacts the bound, making it non-competitive. As for the decomposition on the CIFAR100 experiments, it is interesting to see that the bound from Zakerinia et al. and the one from PB SCH have a similar decomposition, whereas SCH$_+$, despite being penalized by the message probability, relies on a better treatment of its error and confidence term to obtain best bound of the four considered algorithms. This is empirical evidence of the tightness of our bounds compared to those of the runner-ups, all factors (error, confidence $\delta$, …) being kept equal, thanks to the non-linear comparator function $\Delta$ (see Theorem 2.1).
>
> **3. Essential References Not Discussed**
>
> We agree that discussing and conceptually comparing prior work on hypernetworks will enrich the paper. We thank the reviewers for the references from the federated learning literature. They are great examples of the benefits of meta-learned representations and hypernetworks. On a similar note, kindly point toward “Rebuttal KTTB, points 2. and 6.” where the differences between our approaches and model compression approaches.
>
> **4. Questions For Authors**
>
> Concerning the large bound values, many bounds mathematical expressions can give values greater than one. This is especially true when the bound is linear, i.e., of the form “true error $\leq$ empirical error + complexity term” (which is the case for many benchmarks) since the complexity term is usually not upper-bounded. The calculated bound values for Amit & Meir (2018) in Table 2 are truly around 1000, for the complexity term here is too important. We point to the fact that all of our bounds enjoy being bounded in the interval [0,1], so that no vacuous (trivial, >1) bound can be outputted.

---

> > ### Comment · Reviewer_5RkH · 2025-04-03
> >
> > Thanks for your response. I'm still confused about the experiments, and the bounds reported.
> >
> > The bounds from prior works in the paper are meta-learning bounds, and are computed during the meta-training phase. I.e. If there are n training tasks with datasets S_1, ..., S_n the bounds are obtained from S_1, ..., S_n for the expected performance of a future task with dataset S (So a two level generalization bound). However, the bounds in this paper, are bounds based on S, given the results of the meta-learning. These quantities are not comparable. Can you clarify what are the reported bounds for the baselines?
> >
> > ---------------------------------------------------------------
> >
> > Update: Thanks for providing the source code. I checked the code for (Guan & Lu, 2022), and it seems the reported numbers are for meta-test tasks and not the meta-learning bounds. Therefore, based on the code the comparison in the paper is valid (I didn't find the similar part in the second source, but I assume you did the same). However, I strongly suggest to make the differences clear in the paper, and also make your results reproducible by providing the implementation. I increase my score.

---

> > > ### Author Response · Authors · 2025-04-04
> > >
> > > We apologize for the confusion concerning the reported bounds for the baselines. You are right that “If there are $n$ training tasks with datasets $S_1, ..., S_n$ the bounds are obtained from $S_1, ..., S_n$ for the expected performance of a future task with dataset $S$”, while our bounds depend on observations made on a meta-test task. More precisely:
> > > - For (Pentina & Lampert, 2014), (Amit & Meir, 2018), and (Guan & Lu, 2022) benchmarks, we used the implementations of the learning algorithms and bound computations by (Guan & Lu, 2022) (see https://proceedings.mlr.press/v162/guan22b.html, Related Material) ;
> > > - For (Rezazadeh, 2022) and (Zakerinia et al., 2024), we used the implementation we used of the learning algorithms and bound computations by (Zakerinia et al., 2024) (see https://github.com/hzakerinia/Flexible-PAC-Bayes-Meta-Learning/). For example, the bounds reported for "(Guan & Lu, 2022) - kl" and "(Guan & Lu, 2022) - Catoni" correspond respectively to Theorem 3 and 4 in their article, while the bound reported for "(Zakerinia et al., 2024)" corresponds to Theorem 3.1 in their article.
> > >
> > > You are also correct that our bounds and those of the baselines above are not directly comparable, as our methods provide a generalization bound for each specific downstream predictor outputted by our proposed hypernetwork (once a new dataset $S$ is observed). Still, the goal is to certify the result of a meta-learning process, and we reported all bound values to provide comparison points (we are not aware of other works that provide “generalization bounds via meta-learned model representations” as we did). As written above, we will make sure to emphasize this to avoid misconceptions. We will also detail the baseline bounds computation in the appendix.

---

### Official Review · Reviewer_sASN · 2025-03-14

**Overall Recommendation:** 4

**Summary:**

The paper provides novel PAC-Bayesian bounds for meta-learning within the sample compression framework. The approach is based on the hypernetwork architecture. The paper also provides an experiment to show that the proposed bounds can be tighter than prior works. The key technical innovation is extending the sample compression bound (Bazinet et al., 2024) into a setting with real-valued messages.

### update after rebuttal
I am maintaining the current score  following the rebuttal.

**Claims And Evidence:**

The claims made in the submission are well supported.

**Essential References Not Discussed:**

N/A

**Experimental Designs Or Analyses:**

Yes, the experiment setup looks correct to me.

**Methods And Evaluation Criteria:**

Yes.

**Other Comments Or Suggestions:**

1. A typo on line 53 'a'
2. It would be nice to provide real-world examples of "message" in the preliminary section.
3. Line 223, what is $\mu$, in the equation (5) ?
4. For the computed bounds, it would be helpful to include a short description that explains how much better the bounds are compared to prior work, since the bounds are rather complicated and it is not easy to interpret the results.

**Other Strengths And Weaknesses:**

1. The hypernetwork framework is rather complicated. It would be helpful if the authors considered simpler alternative frameworks and included a discussion explaining why each component is important to their approach.

2. The experimental results show mixed performance - good on CIFAR but not ideal on the MNIST pixel swap task. Including more experiments would be more convincing in demonstrating that the proposed method works effectively across different scenarios.

**Questions For Authors:**

1. The bound computation on page 5 is for a uniform distribution on J and the message. The choice is quite simple but seems arbitrary. What kind of distribution would lead to a better bound compared to a uniform one?

2. Regarding Table 1, the best results are bolded. For the bound column, is the bolded value the one with the smallest gap with the true test error?

**Relation To Broader Scientific Literature:**

The key contribution (Theorem 2.4) extends the prior bound for sample compression (Bazinet et al., 2024) to the setting where messages are real-valued.

**Theoretical Claims:**

The theorems in the main text look correct to me.

---

> ### Author Rebuttal · Authors · 2025-04-01
>
> We thank the reviewer for his careful reading of the paper.
>
> **1. Other Strengths And Weaknesses**
>
> **1.1.** Concerning the complexity of the proposed framework, the encoder-decoder architectures are central to our contributions; each component has its unique role in the whole. We are open to performing new experiments if the reviewer has a suggestion on simplifications. That said, we suggest extending the current Appendix G, accompanying Figures 7 to 9 with a comprehensive description and motivation of every choice we made concerning the encoder-decoder architecture.
>
> **1.2.** Concerning the mixed empirical performances, we report empirical evaluation in unfavorable (Table 1) and favorable (Table 2) environments to investigate a new family of methods honestly. Nevertheless, we agree that adding more experiments could reinforce the encouraging performances on the meta-learning binary variants of MNIST and CIFAR100.
>
> That is why we crafted new heatmaps (see https://imgur.com/a/r45Wq56) depicting the test error and generalization bound for PB SCH as a function of both the compression set size and the message size (Recall that Tables 1 and 2 report the performances of the models obtaining the best validation error). These detailed results help in grasping the inner workings of our proposed approach. They depict that using a large message is better-suited for minimizing the test error, but a trade-off between the compression set size and the message size is required to obtain the best bounds. Interestingly, when the message size is restricted to be small, we clearly see the benefit of using $c>0$.
>
> **2. Other Comments Or Suggestions**
>
> Many thanks for pointing out the typos and ambiguities. We will give an example of “message” in Section 2.2 and explain our bounds compared to the ones from the literature in the appendix.
>
> **3. Questions For Authors**
>
> **3.1.** Concerning the choice of a prior distribution over the compression set and the (discrete) message, one can think of using a prior that gives more importance to smaller compression sets and shorter messages. In our current approach, these values are hyperparameters that are fixed, but as discussed in 1.2, the elaboration of an algorithm that selects these quantities as parameters would benefit from such priors. These are interesting options that we will mention as future research directions.
>
> **3.2.** In our tables, the bolded values correspond to the smallest bound values, not those with the smallest gaps.

---

### Decision · Program_Chairs · 2025-05-01

**Decision:**

Accept (poster)

**Comment:**

This paper proves new PAC Bayes sample compression bounds in a meta-learning setting. The main idea is to extend results in the compression literature which apply to the passing of a single message $\omega$ to the case where a prior and posterior distributions are defined on the messages as well.

Although reviewer XmYZ complains about the novelty, the reviewer **agrees that the results are sound**. The other reviewers conceed that the proofs techniques are "heavily inspired" from the literature, but are applied in a novel way which deserves to be discussed and included in the conference.


[1] Marchand, M. & Sokolova, M. Learning with decision lists of data-dependent features. JMLR, 2005.

[2] Bazinet, M., Zantedeschi, V. & Germain, P. Sample compression unleashed: New generalization bounds for real valued losses. AISTATS 2025


[3] Viallard, P., Germain, P., Habrard, A. & Morvant, E. A general framework for the practical disintegration of PAC-Bayesian bounds. Mach. Learn., 2024.